# Structure, catalysis, chitin transport, and selective inhibition of chitin synthase

Dan-Dan Chen[1,2], Zhao-Bin Wang[1,2], Le-Xuan Wang[1], Peng Zhao[1], Cai-Hong Yun [1] ✉ & Lin Bai [1] ✉

Chitin is one of the most abundant natural biopolymers and serves as a critical structural component of extracellular matrices, including fungal cell walls and insect exoskeletons. As a linear polymer of β-(1,4)-linked N-acetylglucosamine, chitin is synthesized by chitin synthases, which are recognized as targets for antifungal and anti-insect drugs. In this study, we determine seven different cryo-electron microscopy structures of a *Saccharomyces cerevisiae* chitin synthase in the absence and presence of glycosyl donor, acceptor, product, or peptidyl nucleoside inhibitors. Combined with functional analyses, these structures show how the donor and acceptor substrates bind in the active site, how substrate hydrolysis drives self-priming, how a chitin-conducting trans-membrane channel opens, and how peptidyl nucleoside inhibitors inhibit chitin synthase. Our work provides a structural basis for understanding the function and inhibition of chitin synthase.

Chitin is the second most abundant natural polysaccharide on earth after cellulose, and ~100 billion tonnes of chitin are produced by living organisms every year[1]. Chitin is a primary component of fungal cell walls, the exoskeletons of crustaceans and insects, and it plays an essential role in the reproduction, growth or development of these organisms[2]. Chitin is a long-chain polymer of β-(1,4)-linked N-acetylglucosamine (GlcNAc) and is synthesized by chitin synthase in the plasma membrane[3,4]. Chitin synthase catalyzes the formation of β(1→4) glycosidic linkages in chitin by using UDP-activated GlcNAc (UDP-GlcNAc) as the sugar donor and meanwhile transports the polysaccharide product out through the membrane (Fig. 1a).

In *Saccharomyces cerevisiae*, chitin synthase is primarily encoded by *CHS1, CHS2* or *CHS3*. Simultaneous knockout of all three genes is lethal in yeast[5]. ScChs1 and ScChs2 belong to the same chitin synthase family, which contains different numbers of transmembrane helices from ScChs3[6]. ScChs1 and ScChs2 are responsible for synthesizing the primary septum and related to cell separation in cytokinesis, while ScChs3 is responsible for the formation of chitin in the bud ring and chitin dispersed in the cell wall[5,7–11]. Previous studies indicated that Chs1 is mainly in a zymogen form and can be activated by proteolysis[3,12,13].

Chitin synthase belongs to the GT-A fold-containing inverting glycosyltransferase 2 family, which also includes hyaluronan and cellulose synthases[14,15]. In recent years, structures of hyaluronan and cellulose synthases have been reported, providing structural insights into their working mechanisms[16–21]. However, the structure of chitin synthase has not been determined, hindering a mechanistic understanding of the catalytic and chitin transport mechanisms of chitin synthase.

As chitin is not available in plants and vertebrates, the biosynthesis of chitin is considered an attractive target for fungicides, insecticides, acaricides, and antifungal drugs[4,22]. For instance, with structural similarity to UDP-GlcNAc, polyoxin B (PolyB) is a peptidyl nucleoside competitive inhibitor of chitin synthase and has been widely used as an antifungal agent against phytopathogenic fungi and arthropod pests in agriculture and forestry for decades[23,24]. Nikkomycin Z (NikkoZ), another peptidyl nucleoside inhibitor of chitin synthase, has shown significant clinical benefits in mammals against pathogenic fungi[25,26]. Determining the structures of chitin synthase in complex with these inhibitors will provide insights into the inhibitory mechanism in atomic level of chitin synthase by peptidyl nucleoside inhibitors, and is also very helpful for development of new antifungal drugs.

In this report, we determined seven cryo-electron microscopy (cryo-EM) structures of *S. cerevisiae* Chs1 in the apo state and in

---

[1]Department of Biochemistry and Biophysics, School of Basic Medical Sciences, Peking University, Beijing, China. [2]These authors contributed equally: Dan-Dan Chen, Zhao-Bin Wang. ✉e-mail: yunch@hsc.pku.edu.cn; lbai@pku.edu.cn

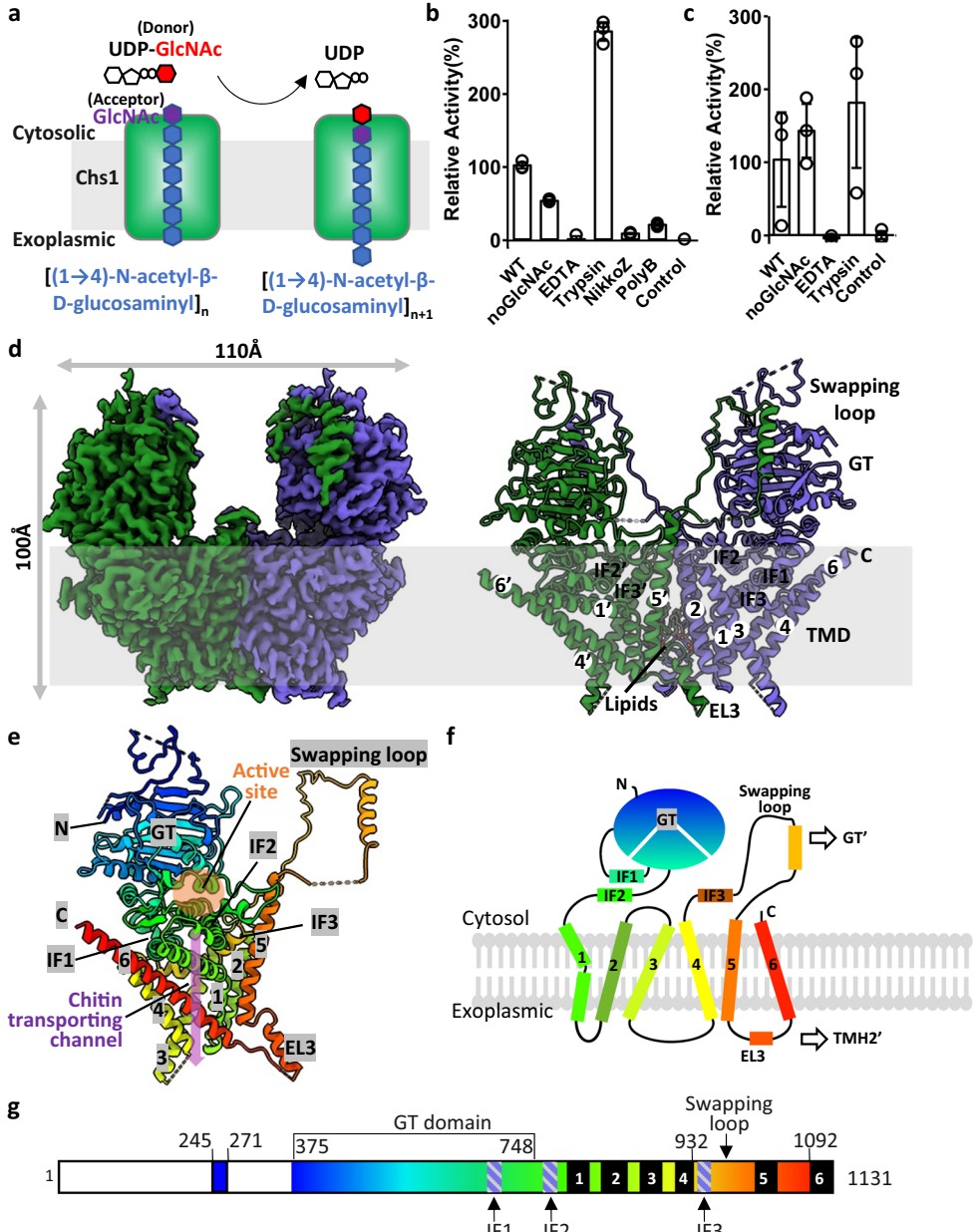

**Fig. 1 | Enzyme activity and cryo-EM structure of Chs1. a** Synthesis and transport scheme of chitin by Chs1. **b** In vitro UDP-Glo glycosyltransferase assay with purified Chs1 in buffer with (WT) or without (noGlcNAc) GlcNAc. The assay under WT conditions was also performed with EDTA, trypsin, NikkoZ, or PolyB. The reaction without Chs1 was used as the control. Data points represent the mean ± SD in triplicate. Source data are provided as a Source Data file. **c** In vitro chitin synthesis assay with purified Chs1 in buffer with (WT) or without (noGlcNAc) GlcNAc. The assay under WT conditions was also performed with EDTA and trypsin. The reaction

without Chs1 was used as the control. Data points represent the mean ± SD in triplicate. Source data are provided as a Source Data file. **d** Cryo-EM map and atomic model of the apo Chs1 dimer. **e** Cartoon representation of the Chs1 monomer. The polypeptide is shown in rainbow from the N- to C-terminus. The putative active site is highlighted by a orange circle. The putative chitin trans-porting channel is outlined by a purple arrow. **f** Cartoon topology of the Chs1 monomer. **g** The Chs1 domain map. Major domains and motifs are labeled. The invisible N-terminal region is in white.

complex with UDP-GlcNAc, UDP-GlcNAc+GlcNAc, UDP, UDP+GlcNAc, PolyB, and NikkoZ. Our studies provide molecular insights into how donor and acceptor substrates bind in the active site, how chitin synthesis is initiated, how the chitin-conducting transmembrane channel works, and the molecular mechanism of how peptidyl nucleoside inhibitors inhibit chitin synthase.

## Results

### Purification and characterization of Chs1

Chs1 was overexpressed in *S. cerevisiae* with a C-terminal triple FLAG tag and purified with anti-FLAG resins followed by size-exclusion

chromatography (Supplementary Fig. 1a, b). The detergent lauryl maltose neopentyl glycol (LMNG) and cholesteryl hydrogen succinate (CHS) were used to stabilize the membrane protein. Previous studies using cell lysates directly or crude purified samples suggested that Chs1 is inactive in vitro and could be activated by proteolysis[3,12,13]. To verify this proposition, we performed an in vitro UDP-Glo glycosyl-transferase assay and chitin synthesis assay using purified Chs1 (Fig. 1b, c). After incubating purified Chs1 or trypsin digested Chs1 with UDP-GlcNAc for 60 min, we measured the free UDP in the reaction solution with a UDP-Glo kit or detected the chitin product by the WGA (wheat germ agglutinin)-coupled immuno HRP method[27]. WGA is a dimeric

lectin that exhibits high binding affinity for GlcNAc residue or oligomers. Consistent with previous studies, the results of both assays revealed the activity of Chs1 was enhanced by trypsin proteolysis with ~1.5–3 times. To our surprise, we also detected some low activity for the purified Chs1 alone (Fig. 1b, c), which was further confirmed by an additional in vitro chitin synthesis assay by staining synthesized chitin with calcofluor white (CFW) (Supplementary Fig. 2a, b). CFW is a non-specific fluorochrome stain for 1,3-β and 1,4-β-linked polysaccharides and displays fluorescence when exposed to long wavelength ultraviolet[28–30]. Besides, the UDP-GlcNAc hydrolysis activity of Chs1 in a buffer with or without EDTA revealed that Chs1 activity is $Mg^{2+}$ dependent (Fig. 1b, c).

Apparently, the detecting activity of our purified Chs1 is against previous studies that wild type Chs1 is inactive. One plausible explanation is that the purified sample had been digested by endogeneous protease during purification and thus activated. To confirm this hypothesis, we performed new purification of Chs1 with excess protease inhibitors throughout the purification (Supplementary Fig. 1a, c). Remarkably, the new purified Chs1 is a mixture of two proteins as shown in SDS-PAGE. Anti-flag western-blot analysis showed both of the two proteins belongs to Chs1, while the larger one is in same size as wild type Chs1 in membrane (Supplementary Fig. 1d). Further tryptic digestion mass spectrometry for the two proteins indicated that the larger protein is indeed wild type Chs1, while the smaller protein lacks about N-terminal 150 residues of Chs1 (Supplementary Fig. 1e, f). Taking together, these results indicated purified Chs1 was partially digested in N-terminal region by endogeneous protease, and thus was partially activated.

To understand how trypsin proteolysis activates Chs1, we incubated purified Chs1 with trypsin in different mass ratios and times, and analyzed the product by SDS-PAGE and tryptic digestion mass spectrometry (Supplementary Fig. 3a–c). We showed proteolysis of purified Chs1 by trypsin generated a product with molecular weight at ~35–40 kDa less than wild type Chs1 (Supplementary Fig. 3a, b), and first ~340 residues in N-terminus of Chs1 is likely removed by trypsin treatment (Supplementary Figure 3c). To further confirm this finding, we expressed and purified a truncated Chs1 by deleting N-terminal 340 residues (Chs1-ΔN). We found Chs1-ΔN is about in the same size as the digested product of purified Chs1 by trypsin (Supplementary Fig. 3a), and was not sensitive to trypsin proteolysis as earlier purified sample. The activity of Chs1-ΔN is also similar with that of trypsin treated wild type Chs1 (Supplementary Fig. 3d). The finding is consistent with our cryo-EM structure of Chs1, in which first ~380 residues of Chs1 are largely disordered and thus more sensitive to trypsin proteolysis (Fig. 1d–g). This finding is also consistent with a previous study that deletion of the non-homologous N-terminal region of both Chs1 and Chs2 had little effect on the trypsin activated enzymatic activity of the corresponding synthase[31]. Taking together, our results revealed that proteolysis activation of Chs1 zymogen is to remove the N-terminal region (NTR), and the N-terminal region of Chs1 zymogen likely plays an inhibitory function.

## The overall structure of Chs1

We performed single-particle cryo-EM analysis on purified Chs1 directly and samples incubated with UDP-GlcNAc, UDP-GlcNAc +GlcNAc, UDP, PolyB, and NikkoZ. Cryo-EM 2D averages showed that Chs1 presented a dimeric architecture, in agreement with the elution volume in the gel filtration (Supplementary Fig. 1a, c, d). Finally, we obtained eight cryo-EM 3D maps of Chs1 in seven different states in a resolution range of 2.4 Å to 3.6 Å (Supplementary Figs. 4–10, Supplementary Table 1).

Using AlphaFold2 predicted structure as initial model, we built the atomic model of Chs1 in apo state into the cryo-EM map at 2.6 Å resolution (Fig. 1d–f, Supplementary Fig. 5). Except for the N-terminal region and a few short loops, Chs1 is mostly ordered and well resolved

(Fig. 1d–f, Supplementary Fig. 11). We also built atomic models for the six remaining 3D maps of Chs1, which are at slightly lower resolutions. All these models were refined to good statistics (Supplementary Table 1) and fit well with the 3D maps (Supplementary Fig. 12). By comparing the 3D maps of Chs1 in the apo, UDP-GlcNAc bound, UDP-GlcNAc+GlcNAc bound, UDP bound, UDP+GlcNAc bound, PolyB bound and NikkoZ bound states, we were able to model corresponding substrates and inhibitors, which all had clear densities in the active site (Supplementary Fig. 13).

Yeast Chs1 has 1131 residues (Fig. 1g). The overall structure of Chs1 reveals a dimer architecture with a twofold symmetry and is ~100 Å tall and 110 Å wide (Fig. 1d). Each Chs1 monomer is composed of a large cytosolic soluble region and a transmembrane domain (TMD), with both the N-terminus and C-terminus in the cytosol. Except a short loop ranging from 245 to 271 residues, the first 374 residues of Chs1 are largely disordered and invisible in our cryo-EM maps. Following the flexible N-terminus is the glycosyltransferase domain (GTD), which is in a conserved GT-A fold with a 10-stranded β-sheet sandwiched by several α-helices and a few β-sheets. In the Chs1 dimer, the two GTDs do not interact directly and likely work independently. The transmembrane domain of Chs1 contains 6 transmembrane helices (TMHs) and is characterized by a kinked TMH1 (Fig. 1e, f). The TMHs are mostly connected by short loops except for a cytosolic domain swapping loop (T969 to T1024) between TMH4 and TMH5 and an extracellular loop (EL3) connecting TMH5 and TMH6. Notably, both the swapping loop and EL3 participate in the formation of the Chs1 dimer interface (described below). GTD interacts with the TMD mainly via three amphipathic interface helices (IF1–3). Among them, IF1 (P645 to F668) and IF2 (V750 to S780) are proximal to TMH1, while IF3 (Q934 to N963) is inserted between TMH4 and TMH5. A large pocket corresponding to the putative catalytic site is presented between GTD and TMD. Below it is a large inside pocket in the transmembrane region, which corresponds to the putative chitin transporting channel (Fig. 1e).

## Dimeric interfaces of Chs1

There are mainly three protein–protein interfaces in the Chs1 dimer: tight packing by TMH2 and TMH5 of both protomers, the interaction between EL3 on the exoplasmic side and TMH2' of the neighboring protomer, and the interaction of the swapping loop with GTD' of the other protomer (Fig. 2a–c). Specifically, TMH2 and TMH5 interact with TMH2', mainly by hydrophobic interactions, and form an inverted "V" shape, which generates a sizable cavity on the exoplasmic side of the membrane (Fig. 2b). Interestingly, we found that two phospholipid molecules fully filled this cavity and were sealed inside by EL3s. The acyl chains of the two lipids are all bent, forming extensive hydrophobic interactions with TMH2 and TMH5. The head and phosphate groups of lipids directly hydrogen-bond to the side chains of R871 in TMH5 and T1061 and D1066 in EL3. Clearly, lipids play a structural role in stabilizing the structure of the Chs1 dimer. The 56 residue-long swapping loop of Chs1 contributes to the dimer interface via extensive contacts with GTD' of the other Chs1 protomer (Fig. 2c). This interface is characterized by many hydrophobic interactions via L979, I982, V994, I1001, L1009, V1011, L1012, and two tyrosine (Y1005 and Y1008) in the swapping loop. Two polar residues (Q1002 and N1004) and segment 989-992 of swapping loop also participate in the interface. Notably, this segment forms a beta sheet with a N-terminal segment 376–379 of GTD'. Most of these residues are conserved (Supplementary Fig. 14).

## Regulatory mechanism of the domain swapping loop

The densities of the swapping loop in our determined structures are relatively weaker than those in other parts of Chs1 or even invisible, indicating that it is only partially stabilized (Fig. 1d, Supplementary Figs. 4–10, 15a, b). It's built by using Alphafold2 predicted structure as the initial model. Such dynamism of the swapping loop may be

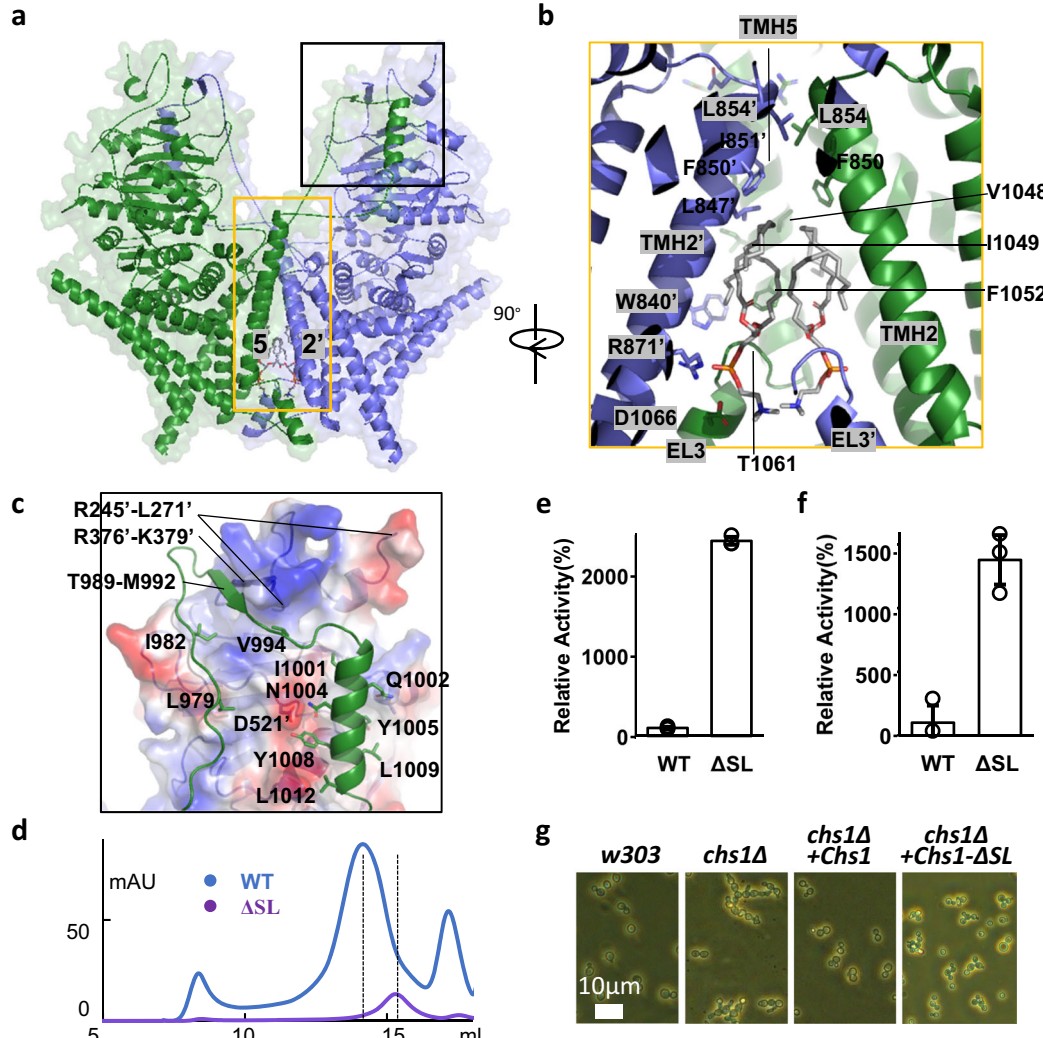

**Fig. 2 | Dimeric interfaces of Chs1. a** Cartoon representation of the Chs1 dimer. The dimeric interfaces are highlighted by black and orange rectangles. **b** An enlarged view of the interface region in the orange box in (**a**). The structure of Chs1 is shown in a cartoon. Key residues and two sealed phospholipid molecules are shown in stick presentation. **c** An enlarged view of the interface region in the black box in (**a**). The domain swapping loop (SL) of one protomer is shown in cartoon representation, and the other protomer is shown in surface electrostatics. Featured residues of SL in the interface are shown in stick. **d** Gel filtration profile of the swapping loop truncated Chs1 (ΔSL) and wild type Chs1 (WT). **e** In vitro UDP-Glo glycosyltransferase assay of Chs1-ΔSL. Data points represent the mean ± SD in triplicate. Source data are provided as a Source Data file. **f** In vitro chitin synthesis assay of Chs1-ΔSL. Data points represent the mean ± SD in triplicate. Source data are provided as a Source Data file. **g** Growth complementation of wild-type w303 cells (w303) and *chs1Δ* cells with empty plasmid (*chs1Δ*) or plasmid carrying either wild-type Chs1 (*chs1Δ*+Chs1) or Chs1-ΔSL (*chs1Δ*+Chs1-ΔSL). Cell images were captured when cells grew to OD = 1. The phenotype of the cell string is an indicator of how much Chs1's function is disrupted in vivo. Three independent experiments were conducted with similar results.

relevant to Chs1's function. To examine the exact role of the swapping loop, we expressed and purified a swapping loop truncated Chs1 (Chs1-ΔSL) in a Chs1 knockout strain, which avoided forming a heterodimer of endogenous wild-type Chs1 and Chs1-ΔSL. The elution volume of gel filtration peak and the native gel demonstrated that purified Chs1-ΔSL is much smaller than wild type Chs1 (Fig. 2d, Supplementary Fig. 15c). SDS-PAGE gel of purified Chs1-ΔSL showed the N-terminal region has been largely degradated during purification as processed by trypsin (Supplementary Fig. 15d), indicating the N-terminal inhibitory region of Chs1-ΔSL is more sensitive to proteolysis. This finding is consistent with our cryo-EM map of Chs1 in apo state, which showed the segment 989-992 of swapping loop packs on segments 245-271 and 376-379 of NTR (Fig. 2c, Supplementary Fig. 15b). Conceivably, removal of swapping loop will absolutely disrupt the interaction and make the NTR more flexible for proteolysis.

We further performed activity assays for Chs1-ΔSL, and found dramatically increased activity of Chs1-ΔSL, which was more than -10

times that of our purified Chs1 (Fig. 2e, f, Supplementary Fig. 2a, b). Such high activity is unexpected because trypsin proteolysis only enhances the activity of Chs1 by 1.5–3 times as mentioned above. It means the increased activity by swapping loop deletion is not only caused by the proteolysis of NTR. Moreover, it's likely that the swapping loop is directly involved in restricting the activity of Chs1 and plays the major inhibitory effect on Chs1 activity. If so, the proteolysis activation of Chs1 may actually be achieved by enhancing the flexibility of swapping loop.

As Chs1-ΔSL was highly active in vitro, we wonder whether Chs1-ΔSL can replace the function of dimeric Chs1 in vivo or not. We performed an in vivo growth complementation assay of Chs1-ΔSL using the yeast Chs1Δ strain (Fig. 2g, Supplementary Figs. 15e, 15). Previous studies indicated that the Chs1Δ strain forms a cell string because of the disrupted building of the primary septum that separates mother and daughter cells. We found that this deficiency can be complemented by a plasmid carrying the wild-type Chs1 gene. In contrast,

Chs1-ΔSL was unable to fully rescue the chs1Δ yeast phenotype like the wild-type Chs1, indicating the swapping loop is essential for regular cellular function of Chs1. Disrupted dimeric state or excessive activity of Chs1-ΔSL are possible reasons for this. It's also possible that deletion of SL damages the localization of Chs1 in plasma membrane.

Besides, we notice that the swapping loop starts with a conserved WGTKG motif (Supplementary Fig. 15a), and mutation of corresponding residues in yeast Chs2 has been proven to be essential for chitin synthesis activity[32]. We also found that both W969A and G970A were unable to rescue the chs1Δ yeast phenotype (Supplementary Fig. 15e, 15), indicating the important role of the WGTKG motif.

**UDP-GlcNAc donor and GlcNAc acceptor binding sites of Chs1**

The cytosolic GT domain of Chs1 adopts a conserved GT-A fold, with the putative catalytic site in the pocket between GTD and TMD. To understand the sugar donor and acceptor binding mechanism, we determined the structure of Chs1 in complex with both UDP-GlcNAc donor and GlcNAc acceptor (UDP-GlcNAc+GlcNAc bound state) at 2.9-Å resolution (Fig. 3a, Supplementary Fig. 6). Comparison of this map with the map in the apo state revealed clear extra densities in the putative active pocket that correspond to the sugar donor (UDP-GlcNAc), acceptor (GlcNAc), and $Mg^{2+}$ (Supplementary Fig. 13).

Although we used wild-type Chs1, UDP-GlcNAc was not hydrolyzed in our structure, probably because of the low incubation temperature and the low activity of Chs1.

In the structure, the nucleotide base group of UDP-GlcNAc is stabilized by T453, M454, Y455, E457, and K578 at the top of the active site by forming an π-π interaction with Y455 and hydrophobic contact with other residues; the phosphate of UDP-GlcNAc interacts with Q756 and R759 in the conserved QXXRW motif and a $Mg^{2+}$; the GlcNAc moiety of UDP-GlcNAc is located right above IF2 in a bent configuration extending outward from the active site. At the bottom of the active site, Y654 in IF1, W760 in IF2, R718, and a conserved VLPGA motif (residues 673–677) form the GlcNAc acceptor binding site (Fig. 3a, b). The GlcNAc ring is vertically inserted into a slit between the indole ring of W760 and the pyrrolidine ring of P675, forming three parallel rings that are ~4 Å apart from each other. Below the GlcNAc molecule are a group of charged residues that keep GlcNAc in the binding site, such as E653, S657, and K662 of IF1. It should be noted that the acceptor binding site is the gate connecting the active site with the chitin transmembrane transporting channel.

Surprisingly, we noticed that the distance between the hydroxy group in C4 of the GlcNAc acceptor and the C1 of the GlcNAc moiety of UDP-GlcNAc is as far as ~7 Å. As the putative catalytic base residue,

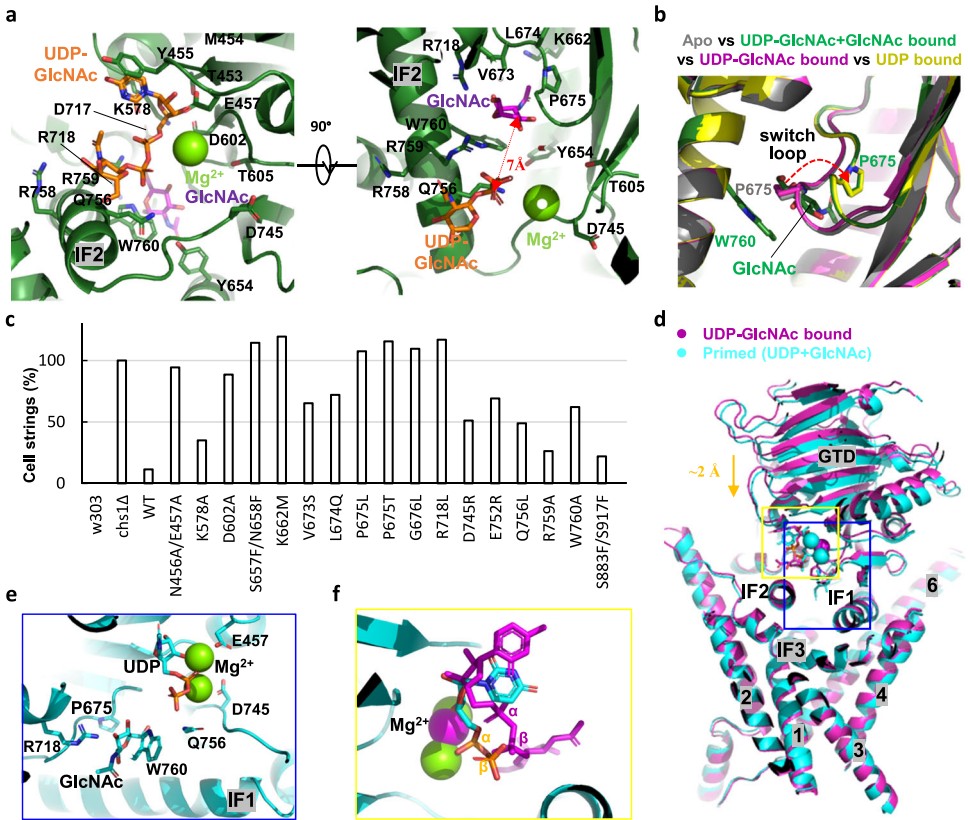

**Fig. 3 | Active site and self-priming of Chs1. a** Close-up view of the active site of Chs1 (green) in complex with UDP-GlcNAc donor, GlcNAc acceptor (purple), and $Mg^{2+}$ (lime). Key residues are shown in stick representation. **b** Structural alignment of Chs1 in apo, UDP-GlcNAc bound, UDP-GlcNAc+GlcNAc bound, and UDP bound states. The red arrow marks the switch loop of Chs1 moving from the apo position to the flipped position. **c** Growth complementation of wild-type w303 cells (w303) and *chs1Δ* cells with empty plasmid (*chs1Δ*) or plasmid carrying either wild-type Chs1 (WT) or mutants. Representative cell images are shown in Supplementary Fig. 13. Cell string refers to a string of unseparated cells with number ≥3. For each of these strains, >2000 cells were counted and calculated using w303 and *chs1Δ* as negative and positive controls. The phenotype of the cell string is an indicator of how much Chs1's function is disrupted in vivo. **d** Structural comparison between UDP-GlcNAc donor-bound Chs1 (purple) and primed UDP+GlcNAc-bound Chs1 (cyan) by aligning their TMs. The orange arrow marks the shift of the GT domain and nucleotide toward the membrane from the donor-bound state to the primed state. The active site is highlighted by a blue rectangle. The sugar donor is highlighted by a yellow rectangle. **e** Close-up view of the active site of primed Chs1 (cyan) in complex with UDP, GlcNAc, and two $Mg^{2+}$ ions (lime). This is enlargement of the blue rectangle in (**d**) viewed from left. Substrates and key residues are shown in stick representation. **f** Enlargement of the yellow rectangle in (**d**) viewed from back. The UDP (cyan) and $Mg^{2+}$ (lime) of primed Chs1, and UDP-GlcNAc (purple) and $Mg^{2+}$ (purple) of the donor-bound Chs1 are shown. α-, β-phosphates are labeled, highlighting a 180° rotation of the β-phosphate from UDP-GlcNAc to UDP.

D717 also does not interact with UDP-GlcNAc in the structure. These findings suggest the GlcNAc moiety of UDP-GlcNAc may need to be rotated horizontally and inserted into active site for the formation of a β(1-4) glycosidic bond in the following reaction (Fig. 3a). The UDP-GlcNAc in our structure is likely in a loading position, and converts to an inserted position for substrate hydrolysis after rotation. In consistent, we found that the GlcNAc moiety of UDP-GlcNAc does not specifically interact with any residue and its density is slightly weaker than that of the UDP moiety, which indicate the GlcNAc group retains some flexibility (Fig. 3a, Supplementary Fig. 13). Furthermore, we found that Mg²⁺, which stabilizes β-phosphate, is only surrounded by E457, D602, T605, and D745 but does not tightly coordinate with any residue in the structure, as their distances are all >4 Å. This is probably partially because the conserved metal coordinate DxD motif in other metal-dependent GT-A fold glycosyltransferases[33] is replaced by a DAG motif (residues 602–604) in Chs1 (Fig. 3a, Supplementary Fig. 14). Such architecture of Mg²⁺ enables the movement of β-phosphate.

We mutated key residues in the active site identified above and performed an in vivo growth complementation assay (Fig. 3c, Supplementary Fig. 16). We found that the N456A/E457A, D602A, and E752R mutants of the donor binding site and the S657F/N658F, K662M, V673S, L674Q, P675L, P675T, G676L, R718L, W760A mutants of the acceptor binding site disrupted Chs1 function significantly, as they were unable to rescue the *Chs1Δ* yeast phenotype. In addition, K578A,

D745R, Q756L, and R759A moderately affected Chs1 function, suggesting that they are less important.

## A switch loop gating the active site to the chitin transporting channel

Structural alignment between Chs1 in the apo- and UDP-GlcNAc +GlcNAc-bound states revealed that most residues in the active site are almost identical except for the conserved VLPGA loop (Supplementary Fig. 17a). In the apo state, the path from the active site to the chitin transmembrane channel is blocked by the VLPGA loop, in which the proline ring of P675 is stabilized by packing with the indole ring of W760 (Figs. 1e, 3b). In contrast, this loop shifted away by 3–5 Å in the UDP-GlcNAc+GlcNAc bound state, resulting in a continuous chitin elongation path from the active site to the inside pocket of Chs1 (Figs. 3b, 4a–d). Movement of the loop from the apo to UDP-GlcNAc +GlcNAc bound state is more like flipping, as the side chains of V673, L674, and P675 are all flipped nearly 180° (Supplementary Fig. 17b). As mentioned above, the growth complementation assay of the P675L, P675T, and G676L mutants revealed their essential roles in Chs1 function (Fig. 3c). Conceivably, flipping of the loop is one of the major steps in the initiation of chitin synthesis by Chs1. Interestingly, Hyaluronan synthase (HAS) has a switch loop (CVGGP) in similar position, but it only slightly shifts up and down during catalysis and does not function as a gate like the loop in Chs1 (Supplementary Fig. 18c, d)[16]. Therefore, we also named the loop of Chs1 "switch loop", which closes

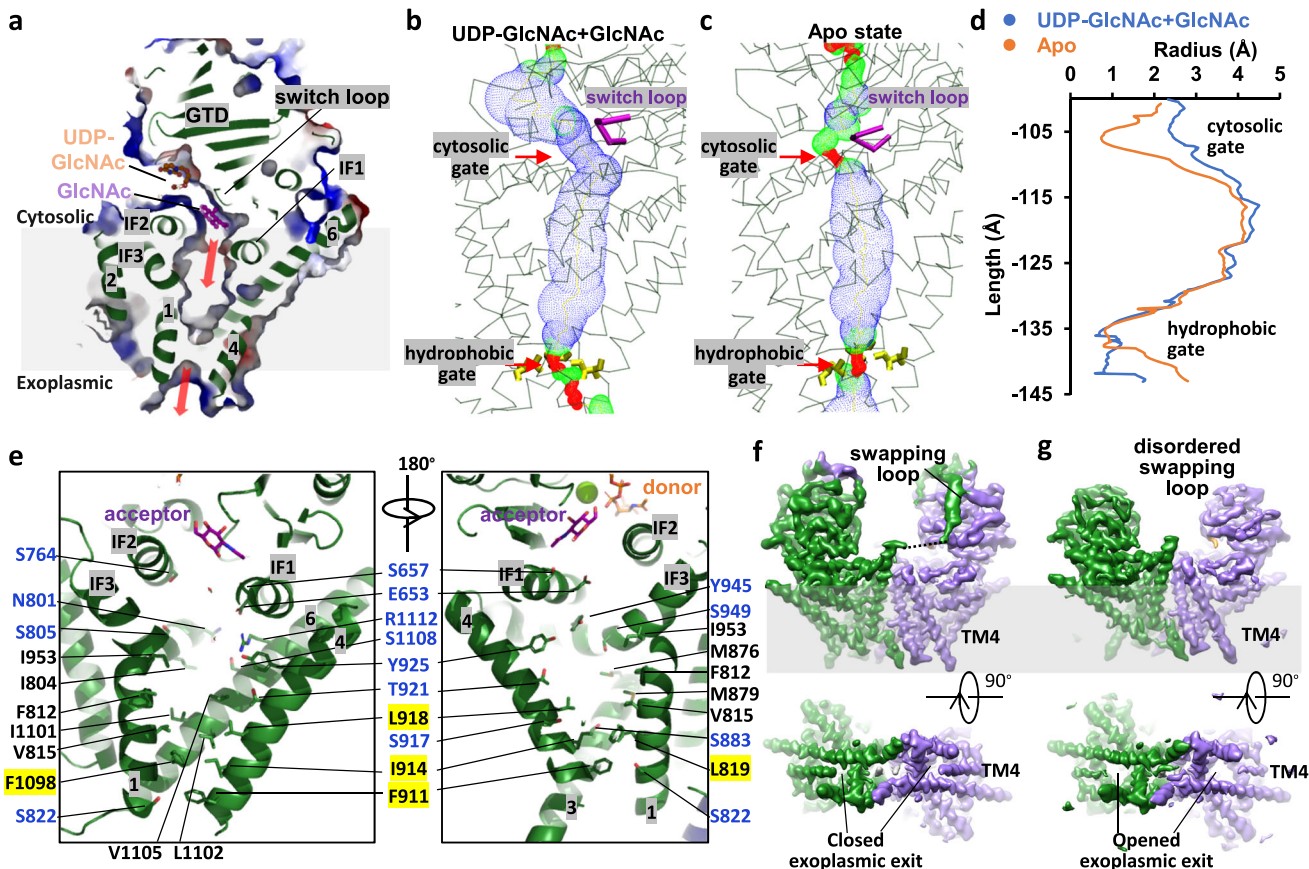

**Fig. 4 | Chitin transporting channel of Chs1. a** Cut-in view of Chs1 (green) in complex with UDP-GlcNAc (orange), GlcNAc (purple), and Mg²⁺ (lime). Chs1 are shown in cartoon and elctrostatic surface. Two red arrows outline the path of chitin transporting channel. Chitin transporting channel of Chs1 in UDP-GlcNAc+GlcNAc bound state (**b**) and apo state (**c**). The channel was calculated by the program HOLE and illustrated by blue dots. The cytosolic gate and exoplasmic hydrophobic gate are highlighted by red arrows. **d** Pore sizes of the chitin transporting pathway in (**b**) and (**c**) are illustrated. **e** Putative transmembrane chitin-transporting channel of Chs1 with residues lining the channel shown as sticks. Labels of polar residue are highlighted in blue. Labels of residues forming the exoplasmic hydrophobic gate are highlighted by yellow background. **f, g** Two extremes of 3DVA analysis of the donor-bound Chs1 structure. The domain swapping loop, TMH4, and exoplasmic exit are labeled in two maps.

the chitin transporting path in the apo state and opens it during chitin synthesis (Fig. 4b–d).

## UDP-GlcNAc hydrolysis drives self-priming and switch loop flipping

Initiation of chitin synthesis requires GlcNAc acceptor binding in the active site of Chs1. Revealing how to generate the first GlcNAc is important to understand the mechanism of chitin synthase. Because of lacking a free GlcNAc source in vivo, previous studies suggest that chitin synthases can "self-prime" by generating GlcNAc from UDP-GlcNAc with an intrinsic UDP-GlcNAc hydrolyzing activity[34,35]. Consistently, we found that Chs1 was active in buffer without GlcNAc (Fig. 1b, c). To unveil the self-priming mechanism of Chs1, we performed time-dependent cryo-EM analysis of Chs1 incubated with UDP-GlcNAc donor for 5 min and 40 min and determined two cryo-EM maps of Chs1 at 3.1-Å resolution (UDP-GlcNAc-10min and UDP-GlcNAc-40min) (Supplementary Fig. 5, 6). According to the densities of the substrates, we found that the UDP-GlcNAc-5 min structure is in a state with UDP-GlcNAc in loading position (UDP-GlcNAc-loading state), while the UDP-GlcNAc-40 min structure is in the UDP+GlcNAc bound state (primed state) (Supplementary Fig. 13). This indicates that Chs1 in the latter structure has a fully hydrolyzed UDP-GlcNAc donor and was self-primed.

In the UDP-GlcNAc-loading structure, the switch loop is in the apo position, indicating that UDP-GlcNAc binding does not activate the switch loop (Fig. 3b, d). In the primed Chs1 structure, the switch loop is in the flipped position, and GlcNAc is sandwiched by P675 and W760 (Fig. 3d, e). The C3 hydroxyl of GlcNAc forms a hydrogen bond with R718, which positions the C4 hydroxyl of GlcNAc toward the donor binding site (Fig. 3e). Furthermore, we noticed that UDP-GlcNAc hydrolysis induces rigid body movement of the GT domain toward the membrane by ~1.5–2.5 Å, which may help to push the product into the channel (Fig. 3d). Similar with the GT domain, the bound UDP also moves toward the membrane, forming a new unique configuration. The β-phosphate group of UDP rotated by nearly 180° toward the acceptor binding site from the UDP-GlcNAc loading state to the primed state (Fig. 3f). The rotated β-phosphate is mainly stabilized by Q756 and R759 (Fig. 3d). Notably, two $Mg^{2+}$ ions were identified to stabilize the α-phosphate by tight coordinating with E457 and D745 in the primed Chs1 structure, which is different from the relative loose binding of $Mg^{2+}$ in the UDP-GlcNAc loading state (Fig. 3d, e). These findings support our earlier suggestion that the GlcNAc moiety of UDP-GlcNAc needs to be rotated to an inserted positon for the formation of β(1 → 4) glycosidic bonds.

To confirm which product of UDP-GlcNAc hydrolysis induces switch loop flipping, we further determined two cryo-EM maps of Chs1 incubated with GlcNAc or UDP at 3.5 Å and 3.6 Å resolution, respectively (Supplementary Fig. 9). We found that the first structure was still in the apo state without GlcNAc binding, suggesting GlcNAc alone could not activate the switch loop (Supplementary Fig. 13). In contrast, the switch loop was activated by the UDP product alone, as shown in the UDP-bound Chs1 structure (Fig. 3b). The UDP is also in a flipped configuration, similar with UDP in the primed state. As a whole, the results suggest the process of UDP-GlcNAc hydrolysis into UDP, and reconfiguration of UDP induces flipping of the switch loop and opening of the path from the active site to the transmembrane channel.

## The transmembrane chitin transporting channel

The putative transmembrane chitin-transporting channel of Chs1 is mainly composed of TMH1, TMH3-4, and TMH6 and positioned directly below the active site (Fig. 4a, e). These TMHs form a funnel-shaped cavity in the membrane, which features by a hydrophilic top part and a hydrophobic bottom part (Fig. 4a, e). There are numerous polar residues in the top part of the cavity, including E653, S657, and D661 of IF1; N801, S805, and S808 of TMH1; Y925, Y945, and S949 of

IF3; and S1108 and R1112 of TMH6. The hydrophilic environment is more favorable to the chitin elongation and transport. In contrast, the bottom part of the cavity is mainly composed of hydrophobic residues (Fig. 4e). Among them, L819 of TMH1, M886 of TMH3, F911, I914, L918 of TMH4, and F1098 of TMH6 form a closed hydrophobic gate, blocking the cavity from connecting to the exoplasmic side. Conceivably, the bottom hydrophobic gate needs to be opened during chitin synthesis. Two polar residues close to the hydrophobic gate, S883 and S917, also seem to play important roles during this process, as shown in the in vivo growth complementation assay of the S883F/S917F double mutant (Fig. 3c).

The exoplasmic end of TMH4 and its adjacent loop connecting TMH3 and TMH4 are found to be highly flexible, indicated by their weak densities in all of our cryo-EM maps. This suggests that TMH4 is movable, and the dynamics likely correspond to the hydrophobic gate opening of Chs1 during chitin synthesis. To further confirm this hypothesis, we carried out a 3D variability analysis (3DVA) of the Chs1 structure in UDP-GlcNAc loading state. 3DVA is a newly developed software in cryoSPARC to reveal the continuous variability and discrete heterogeneity of a structure[36]. We found that TMH4 morphs from well-ordered to disordered in two extremes, and accordingly the chitin transporting channel is closed and opened (Fig. 4f, g, Supplementary Movie 1). This confirmed the flexibility of TMH4 in the map. We also observed that the density of the domain swapping loop became disordered simultaneously when TMH4 density became disordered. In this state, Chs1 has an opened chitin translocation channel and don't have domain swapping loop bound in GTD (Fig. 4g). This may suggest that the translocation channel opening is correlated with the release of an inhibitory swapping loop from the GTD domain.

The cavity has a lateral opening in the middle of the lipid bilayer, which was formed by TMH3, TMH4 and IF3 (Supplementary Fig. 19). Furthermore, two horizontal elongated densities are observed filling in the cavity from the lateral opening, and the densities resembled two sterol molecules according to their shapes. It is unclear whether sterol molecules are available in vivo or introduced during purification, but they clearly need to diffuse out of the cavity during chitin transport. Similar lipids were also found in the structure of hyaluronan synthase[16].

## Mechanism of peptidyl nucleoside inhibitors of chitin synthase

Polyoxins and nikkomycins are both natural peptidyl nucleoside antibiotics. With structural similarity to UDP-GlcNAc, PolyB and NikkoZ, two major components of peptidyl nucleoside inhibitors, exhibit antifungal and anti-insect activity by acting as competitive inhibitors of chitin synthase. The nucleoside moieties of PolyB and NikkoZ are the same 5-aminohexuronic acid N-glycosidically bound to uracil. In contrast, the peptidyl moieties of PolyB contain carbamoylpolyoxamic acid, and NikkoZ contains hydroxypyridylhomothreonine. We performed the in vitro UDP-Glo glycosyltransferase assay with PolyB or NikkoZ in the reaction buffer and found that the Chs1 activity was reduced by more than 80% (Fig. 1b, Supplementary Fig. 3d).

We determined the structures of Chs1-PolyB and the Chs1-NikkoZ complex at 3.6-Å and 2.4-Å resolution, respectively (Supplementary Figs. 4, 10). Comparison between Chs1 maps in the apo and inhibitor-bound states revealed clearly elongated extra densities corresponding to PolyB and NikkoZ in the active site, consistent with their function as substrate competitive inhibitors (Supplementary Fig. 13). The nucleoside moieties of PolyB and NikkoZ are stabilized by T453, M454, Y455, and K578 at the top of the active site, which are the same residues interacting with the nucleoside group of UDP-GlcNAc (Fig. 5a–d). In contrast, unlike the flexible GlcNAc moiety of UDP-GlcNAc extending away from the active site, the peptidyl moieties of PolyB and NikkoZ extend toward the chitin transport channel and form extensive interactions with E457, D602, A677, and W760 (Fig. 5a–d). The unique architecture of peptidyl moiety makes the inhibitor locked inside of the active site of Chs1. Specifically, NikkoZ has extra contacts with P675

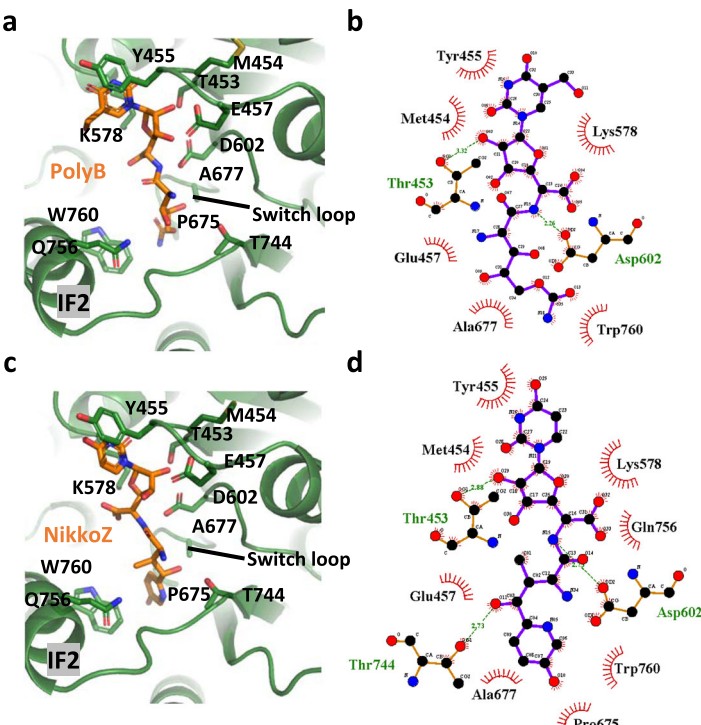

**Fig. 5 | Structural basis for the inhibition of Chs1 by two peptidyl nucleoside inhibitors. a** Detailed binding site of Polyoxin B. Key residues and Polyoxin B are displayed in sticks. **b** LIGPLOT scheme for the Polyoxin B binding residues of Chs1.

**c** Detailed binding site of Nikkomycin Z. Key residues and Polyoxin B are displayed in sticks. **d** LIGPLOT scheme for Nikkomycin Z binding residues of Chs1.

and Q756 of Chs1 because of its larger group in the peptidyl moiety than PolyB (Fig. 5c, d), which is consistent with previous findings that Nikkomycin has a relatively lower Ki than Polyoxins[26,37]. The unique interactions between Chs1 and the peptidyl moieties of PolyB and NikkoZ also indicate the higher binding affinity of these inhibitors over UDP-GlcNAc substrate. Besides, the switch loop of Chs1 is open in our Chs1-PolyB and Chs1-NikkoZ structures. Notably, end part of the peptidyl moieties of PolyB and NikkoZ occupy the GlcNAc acceptor binding site, and thus block the chitin transport channel. Therefore, our structures suggest that the peptidyl nucleoside inhibitors inhibit chitin synthases using a unique mechanism: the nucleoside moiety competes with UDP binding, and the peptidyl moiety induces the switch loop open and blocks the gate of chitin transport channel.

## Discussion

In this work, we first revealed proteolysis activation of Chs1 zymogen is through removal of its N-terminal region. By determining the structure of dimeric Chs1 in apo, donor-loading, primed (UDP+GlcNAc bound), and donor+acceptor bound states, we were able to propose a working model of activated Chs1 (Fig. 6a). In the apo state of Chs1, the putative active site is open for donor binding, whereas the putative chitin transporting channel is sealed by a switch loop in cytosolic side and TMH4 in exoplasmic side. Then, UDP-GlcNAc binds to the active site mainly by its UDP moiety, while the GlcNAc moiety is flexible for self-priming. Following that, donor hydrolysis triggers the flipping of the switch loop and opens the gate from the active site to the transporting channel. Donor hydrolysis generates a GlcNAc molecule in the acceptor binding site with its C4 hydroxyl toward the active site. After that, UDP is released, and the second donor molecule binds the active site, as shown in the UDP-GlcNAc+GlcNAc bound state of Chs1. We further propose that a following substrate binding and turnover generates a β(1 → 4) glycosidic bond with the acceptor GlcNAc to form a disaccharide. Chitin elongates by repeating the process of donor binding and hydrolysis. Finally, the flexible TMH4 shifts outward,

opening the chitin transporting channel. The downward shifting of GTD to the membrane during donor hydrolysis may help to push the product out. Peptidyl nucleoside inhibitors inhibit chitin synthases by occupying both donor and acceptor binding sites and blocking the chitin transport channel (Fig. 6b).

The active site and chitin transporting channel of each Chs1 subunit in the dimer are independent, but the domain swapping loop in one subunit probably regulates the other subunit's function by inhibiting and releasing its GTD. This makes the chitin synthesis of the two subunits simultaneous, which is probably important for forming chitin fibers (Fig. 6c). Besides, because two neighbor GlcNAc residues within the chitin chain are in alternating orientations, previous study proposed that chitin synthase possesses two active sites for each sugar orientation[38]. However, our structure of Chs1 showed each subunit only has one active site and two active sites in Chs1 dimer are too far to cooperate with each other. Further research is required to fully understand the mechanism of chitin elongation of Chs1, including how to connect GlcNAc residues in alternating orientations. Capturing Chs1 in complex with chitin will undoubtedly provide more insights into this model.

Chitin, hyaluronan and cellulose synthases all belong to the GT-2 family. Structural comparison indicated that the Chs1 shares overall similarity with hyaluronan and cellulose synthases (Supplementary Fig. 18a, b)[16,21], although their structural details and oligomeric state are different. Chitin, hyaluronan and cellulose synthases have a similar cytosolic glycosyltransferase domain and a transmembrane polysaccharide transporting channel. Notably, the GlcNAc moiety of the UDP-GlcNAc donor in complex with HAS faces the acceptor binding site (Supplementary Fig. 18c, d), which supports our proposition that the GlcNAc moiety of UDP-GlcNAc in Chs1 needs to be rotated inward for sugar transfer.

During preparation of this manuscript, another group reported the cryo-EM structures of chitin synthase 2 from *Candida albicans* (caChs2) in apo, UDP-GlcNAc bound, nikkomycin Z bound, and

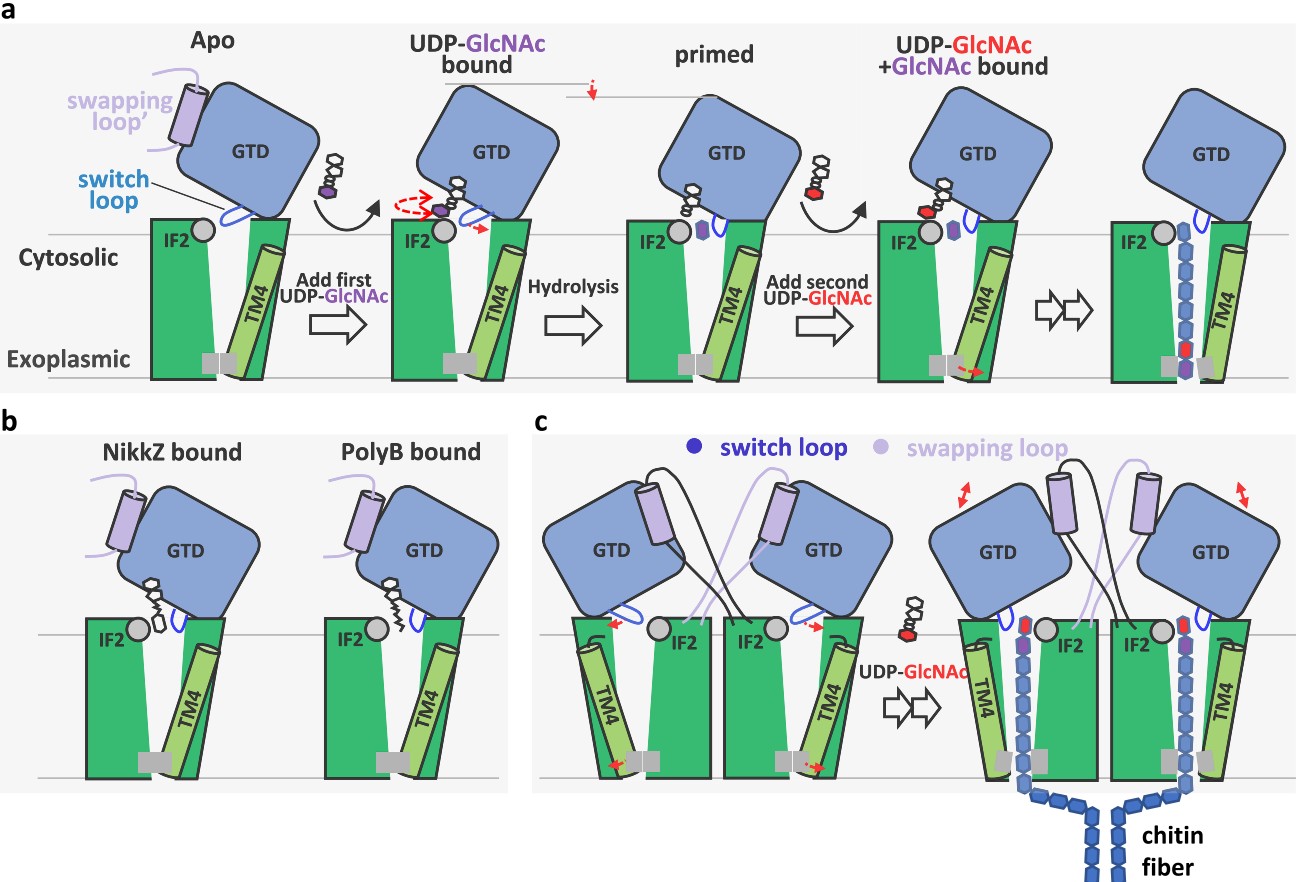

**Fig. 6 | Proposed catalytic and inhibitory mechanism of Chs1. a** Chs1 in the apo state has an open active site and a transmembrane channel sealed by a switch loop and TMH4. The sugar donor (UDP-GlcNAc) binding and subsequent hydrolysis primed the first acceptor molecule (GlcNAc) in the binding site. The priming process opens the chitin transport channel on the cytosolic side by flipping the switch loop and triggers the downward movement of the GT domain to push the product out. Then, UDP is released, and a new donor molecule binds to the active site for chitin elongation. To transport the chitin product outside, TMH4 is shifted outward, resulting in an open channel on the exoplasmic side. More details are provided in the text. **b** Peptidyl nucleoside inhibitors inhibit Chs1 by competing with UDP binding, occupying the sugar acceptor binding site, and blocking the chitin transport channel. **c** The domain swapping loop in one subunit regulates the other subunit's activity by inhibiting and releasing its GTD. This allows the chitin synthesis of two subunits in the Chs1 dimer simultaneously and facilitates chitin fiber formation. During chitin synthesis, the switch loop and TMH4 of each subunit shift to open the chitin transporting channel.

polyoxin D bound states[34]. caChs2 and Chs1 from *Saccharomyces cerevisiae* (scChs1) in our work share sequence identity of ~50%. Structures of caChs2 and scChs1 in four different states are all superimposable with a root-mean-square-deviation (rmsd) of ~1.2 Å, indicating their high similarity (Supplementary Fig. 20). The major substantial difference observed in structural comparison is that the conserved VLPGA motif (switch loop) is closed in apo and UDP-GlcNAc bound scChs1, but opened in apo and UDP-GlcNAc bound caChs2. This difference is probably caused by the species difference or different expression system used in two works (scChs1 in *S. cerevisiae* BY4742; caChs2 in sf9 insect cell). Besides, the UDP-GlcNAc substrate and two types of inhibitors are all bound in similar positions in structures of caChs2 and scChs1, suggesting their similar catalytic and inhibitory mechanisms. Because of the lack of structures in primed and UDP-GlcNAc+GlcNAc bound states, detailed catalytic cycle of chitin synthase and regulation of the chitin transporting channel by TMH4 were not discovered in the work of caChs2. Inhibitory function of the domain swapping loop was also not defined in that work.

In conclusion, our systematic structural and biochemical studies of Chs1 have provided deep insights into the molecular mechanisms for proteolysis activation, glycosyltransferase activity and transmembrane chitin transport of a fungal chitin synthase, and the inhibitory mechanisms of two peptidyl nucleoside chitin synthase inhibitors. Given the critical role of chitin synthase in pathogenic fungal

infections, our work provides a platform for the development of new small antifungal molecules.

## Methods

### Expression and purification of Chs1

We tagged Chs1 (Uniprot ID: P08004) with a C-terminal triple-FLAG to the modified pRS423 vector (Supplementary Table 2). The yeast strain BY4742 was transformed and cultured in synthetic histidine-dropout medium (SD-His) for approximately 20 h and then transferred to YPG medium (10 g yeast extract, 20 g peptone, 20 g D-galactose per liter) for 12 h before harvest. Cells were resuspended in lysis buffer (20 mM Tris, pH 7.4, 0.2 M sorbitol, 50 mM potassium acetate, 2 mM EDTA, and 1 mM phenylmethylsulfonylfluoride (PMSF) and then lysed using a French press at 15,000 psi. We centrifuged the lysate at $10,000 \times g$ for 30 min at 4 °C and collected the supernatant for another centrifuge cycle at $100,000 \times g$ for 60 min at 4 °C. The membrane pellet was collected and then resuspended in buffer A containing 10% glycerol, 20 mM Tris-HCl (pH 7.4), 1% DDM, 0.1% cholesteryl hydrogen succinate (CHS), 500 mM NaCl, 1 mM MgCl$_2$, 1 mM EDTA, and 1 mM PMSF. After incubation for 30 min at 4 °C, the mixture was centrifuged for 30 min at $100,000 \times g$ to remove the insoluble membrane. We loaded the supernatant into a pre-equilibrated anti-FLAG (M2) affinity column (GenScript) at 4 °C and washed the affinity gel with buffer B (20 mM HEPES, pH 7.4, 150 mM NaCl, 0.01% lauryl maltose neopentyl glycol

(LMNG), 0.0033% GDN, 0.0013% CHS, and 1 mM MgCl$_2$). The proteins were eluted with buffer B containing 0.15 mg/mL 3 × FLAG peptide and were further purified in a Superose 6 10/300 Increase gel filtration column in buffer C (20 mM HEPES, pH 7.4, 150 mM NaCl, 0.001% LMNG, 0.00033% GDN, 0.00013% CHS, and 1 mM MgCl$_2$). The purified proteins were assessed by SDS–PAGE and concentrated for cryo-EM analysis. We screened a range of detergents in the purification, and found LMNG/GDN/CHS is the best detergent for preparing cryo-EM grid, and DDM/CHS is the best for stability and yield of Chs1. To make the experiment simple, DDM/CHS purified sample was used in the activity assays.

## Cryo-electron microscopy

To capture different states, purified Chs1 was used for EM grid preparation directly or mixed with different substrates (5 mM UDP-GlcNAc, 5 mM GlcNAc, or 5 mM UDP) or inhibitors (5 mM Nikkomycin Z or 5 mM Polyoxin B) on ice. After incubation, 2.5-µL aliquots of Chs1 at a concentration of ~3 mg/mL were placed on glow-discharged holey carbon grids (Quantifoil Au R1.2/1.3, 300 mesh) and flash-frozen in liquid ethane using an FEI Vitrobot Mark IV. Grids were screened in a 300-keV FEI TF30 electron microscope from the School of Basic Medical Sciences, Peking University. Cryo-EM data were collected automatically with SerialEM in a 300-keV FEI Titan Krios electron microscope from the School of Physics, Peking University and Institute of Biophysics Chinese Academy of Sciences with defocus values ranging from −1.2 to −2.5 µm. The microscope was operated with a K3 direct detector at a nominal magnification of 225,000×. The total doses were 50–60 electrons per Å$^2$ at the sample level.

## Cryo-EM image processing

We used the program MotionCorr-2.0[39] for motion correction and CTFFIND-4.1[40] for calculation of contrast transfer function parameters. We used RELION-3 for particle picking and extraction and used CryoSPARC for all remaining steps[36,41]. The resolution of the maps was estimated by the gold-standard Fourier shell correlation at a correlation cutoff value of 0.143.

For the Nikkomycin Z bound Chs1 structure, we collected 2711 raw movie micrographs. A total of 4,135,734 particles were picked automatically for 2D classification and heterogeneous refinement. Based on the quality of three heterogeneous refined 3D map, 837,015 particles were retained for further refinement, and postprocessing, resulting in a 2.4-Å average resolution 3D map using C2 symmetry.

For the Chs1 structure in apo state, we collected 3672 raw movie micrographs. A total of 3,529,578 particles were picked automatically for 2D classification and heterogeneous refinement. Based on the quality of three heterogeneous refined 3D map, 1,342,168 particles were retained for further refinement, and postprocessing, resulting in a 3.0-Å average resolution 3D map using C2 symmetry.

For the UDP-GlcNAc+GlcNAc bound Chs1 structure, we collected 1252 raw movie micrographs. A total of 996,080 particles were picked automatically for 2D classification and heterogeneous refinement. Based on the quality of three heterogeneous refined 3D map, 208,320 particles were retained for further refinement, and postprocessing, resulting in a 3.1-Å average resolution 3D map using C2 symmetry.

For the UDP-GlcNAc bound Chs1 structure, we collected 1076 raw movie micrographs. A total of 1,036,417 particles were picked automatically for 2D classification and heterogeneous refinement. Based on the quality of three heterogeneous refined 3D map, 330,289 particles were retained for further refinement, and postprocessing, resulting in a 3.1-Å average resolution 3D map using C2 symmetry.

For the UDP+GlcNAc bound Chs1 structure (primed state), we collected 2469 raw movie micrographs. A total of 1,059,472 particles were picked automatically for 2D classification and heterogeneous refinement. Based on the quality of three heterogeneous refined 3D map, 184,449 particles were retained for further refinement, and

postprocessing, resulting in a 3.1-Å average resolution 3D map using C2 symmetry.

For the UDP bound Chs1 structure, we collected 632 raw movie micrographs. A total of 431,546 particles were picked automatically for 2D classification and heterogeneous refinement. Based on the quality of three heterogeneous refined 3D map, 115,773 particles were retained for further refinement, and postprocessing, resulting in a 3.6-Å average resolution 3D map using C2 symmetry.

For the Chs1 structure in apo state after incubation with GlcNAc, we collected 1564 raw movie micrographs. A total of 1,077,654 particles were picked automatically for 2D classification and heterogeneous refinement. Based on the quality of three heterogeneous refined 3D map, 218,698 particles were retained for further refinement, and postprocessing, resulting in a 3.5-Å average resolution 3D map using C2 symmetry.

For the Polyoxin B bound Chs1 structure, we collected 2311 raw movie micrographs. A total of 2,608,153 particles were picked automatically for 2D classification and heterogeneous refinement. Based on the quality of three heterogeneous refined 3D map, 142,887 particles were retained for further refinement, and postprocessing, resulting in a 3.6-Å average resolution 3D map using C2 symmetry.

## Structural modeling, refinement, and validation

We used the predicted Chs1 structure by Alphafold2 as the initial model[42,43], fitted it into our map, and corrected it in COOT[44] and Chimera[45]. The complete Chs1 model was refined by real-space refinement in the PHENIX program[46] and subsequently adjusted manually in COOT. Finally, the model was validated using MolProbity[47]. Structural figures were prepared in Chimera and PyMOL (https://pymol.org/2/).

## UDP-Glo glycosyltransferase assay

The activity of Chs1 was measured using the UDP-Glo™ Glycosyltransferase Assay from Promega (Catalog: V6961), which can measure the activity of any glycosyltransferase (GT) that uses a UDP-sugar as a substrate. Each reaction contained 0.125 µg purified protein, 20 mM HEPES, pH 7.4, 150 mM NaCl, 0.025% DDM, 0.0025% CHS and 10 mM MgCl$_2$, 10 mM GlcNAc and 10 mM UDP-GlcNAc in a total volume of 5 µL. The mixture was first incubated for 60 min at 30 °C, and then 5 µL of UDP Detection Reagent was added. The mixture was incubated for 60 min at room temperature, and then luminescence was detected by a Synergy H1 Hybrid Multi-Mode Microplate Reader (BioTek).

## Chitin synthesis assay

Chitin synthase activity was evaluated using the method described by Lucero with some modification[27]. The assay mainly includes two steps: binding of synthesized chitin to a WGA-coated surface and detection of the polymer with a horseradish peroxidase-WGA conjugate. Horseradish peroxidase activity can be determined in absorbance at 600 nm, and the values are converted to amounts of chitin using commercial chitin as a standard. Firstly, wells of 96-well plates were precoated with WGA (wheat germ agglutinin). Then, 100 µL reaction solution containing 3.4 µg purified protein, 20 mM HEPES, pH 7.4, 150 mM NaCl, 0.025% DDM, 0.0025% CHS, 1 mM MgCl2, 2 mM GlcNAc and 1 mM UDP-GlcNAc was added and incubated for 60 min at room temperature. After the solution in the wells was removed, 100 µL WGA-HRP (wheat germ agglutinin - horse reddish peroxidase) at a concentration of 1 µg/mL in buffer solution (0.2 mM HEPES at pH 7.4) was added to each well and cultured at 37 °C for 20 min. Then, the solution in the wells was removed, and the wells were washed with buffer solution three times. To each well, 100 µL of 1-Step Ultra TMB-ELISA substrate solution was added, and the mixtures were lucifugely reacted for 10 min at room temperature. Finally, all reactions were stopped by 100 µL 0.5 M H$_2$SO$_4$, and UV−Vis absorption spectra were recorded on a microplate reader at

450 nm by a Synergy H1 Hybrid Multi-Mode Microplate Reader (BioTek).

## Tryptic digestion mass spectrometry

The protein bands in SDS-PAGE gel were firstly cutted and processed by in-gel digestion. After distained and reductive alkylated, the gel pieces were incubated with 500 ng trypsin in 100 μL of 50 mM $NH_4HCO_3$ overnight at 37 °C. The trypisn-digested peptides were then eluted from the gel, and dried by vacuum. The nano-LC-MS/MS experiments were then performed using a LTQ orbitrap velos pro mass spectrometer (Thermo Electron,Bremen, Germany). Each sample was loaded into a C18 reverse phase column (Thermo Electron, Bremen, Germany). The peptide mixtures were eluted with a 0-40% gradient (Buffer A, 0.1% formic acid, Buffer B, 0.1% formic acid and 100% ACN) over 60 min and were detected in mass spectrometer using a data-dependent TOP10 method. The general mass spectrometric conditions were: spray voltage, 2.2 kV; no sheath and auxiliary gas flow; ion transfer tube temperature at 250 °C; 35% normalized collision energy using for MS/MS(MS2). Ion selection thresholds were 1000 counts for MS2. An activation $q = 0.25$ and activation time of 30 ms were applied in MS2 acquisitions. The mass spectrometer was operated in positive ion mode and a data-dependent automatic switch was employed between MS and MS/MS acquisition modes. For each cycle, one full MS scan in the Orbitap followed by ten MS2 in the LTQ on the ten most intense ions. Selected ions were excluded from further selection for 30 s. Final LC-MS/MS data were submitted to database searching against the *S. cerevisiae* sequence library in the Uniprot protein sequence database using Sequest HT algorithm in the Proteome Discoverer 1.4 software package (Thermo Scientific).

## Chitinase assay

Insoluble chitin was synthesized following above chitin synthesis assay and was collected by centrifugation. The chitin was washed and resuspended with buffer (50 mM $NaH_2PO_4$, pH 6.0). The chitinase (C6137, Sigma, 0.5 mg/ml) was added and incubated at 37°C for 24 h.

## Yeast growth complementation assay

We prepared the Chs1 knockout strain (ΔChs1) in the W303 strain (Supplementary Table 2). Chs1 mutants and truncations (N456A/ E457A, K578A, D602A, S657F/N658F, K662M, V673S, L674Q, P675L, P675T, G676L, R718L, D745R, E752R, Q756L, R759A, W760A, S883F/ S917F, and Δ975-1024) were constructed using plasmid pRS423-His3 and transformed into the ΔChs1 strain (Supplementary Table 2). The cells were first grown to the same OD value of 1.0 in SD medium at 30 °C. Microscopy was performed with an OLYMPUS CKX53 inverted microscope at a magnification of 400×. Image acquisition and analysis were performed with the program ToupView 3.7. The displayed microscopic images of control and knockout/mutant samples were adjusted equally using the same brightness and contrast values. Yeast cells were briefly washed with water and immediately imaged in water at room temperature.

## Characterization of synthesized chitin by CFW stain

Calcofluor white (CFW) is a non-specific fluorochrome stain for 1,3-β and 1,4-β-linked polysaccharides and displays fluorescence when exposed to long wavelength ultraviolet. The staining of synthesized chitin by CFW was performed by a modification of the previous procedure[28–30]. We prepare 20 μL reaction solution containing 3 μg purified protein, 20 mM HEPES, pH 7.4, 150 mM NaCl, 0.025% DDM, 0.0025% CHS, 1 mM $MgCl_2$, 2 mM GlcNAc and 2 mM UDP-GlcNAc and incubated for 0, 1, and 24 h, respectively at 30 °C. Finally, 10 mM EDTA was added into the reaction solution to stop the reaction. To visualize synthesized chitin, we put 3 μl of the reaction solution on a clean glass slide, then add 3 μl of Calcofluor White Stain and equal volume of 10%

Potassium Hydroxide, cover the specimen with a coverslip and leave it to absorb the stain for 1 minute. Finally, the stained chitin was detected using a Leica DMI4000 B automated inverted research fluorescence microscope under ultraViolet rays at x200-x400 magnification. Image acquisition and analysis were performed with the program Leica Application Suite X.

## Blue Native PAGE

Blue Native PAGE was carried out under nondenaturing conditions in gels (1.0 mm thickness) containing 4–15% polyacrylamide. It was carried out in Blue Native PAGE electrophoresis running buffer (Sangon Biotech (Shanghai) Co., Ltd.) at 100 V for 15 h.

## Western blotting

SDS–PAGE was first performed in gels (1.0 mm thickness) containing 4–12% polyacrylamide. Proteins in SDS–PAGE gel were then transferred to a polyvinylidene fluoride membrane. Anti-Flag mouse monoclonal antibody (Abcam, Catalog: ab125243, Clone: FG4R) and HRP conjugated goat anti-mouse IgG (H + L) (Proteintech, Catalog: SA00001-1) were used for blotting at 1:1000 and 1:5000 dilution ratio respectively.

## Reporting summary

Further information on research design is available in the Nature Portfolio Reporting Summary linked to this article.

## Data availability

The cryo-EM 3D maps and the corresponding atomic models of Chs1 have been deposited in the EMDB database and the RCSB PDB with the respective accession codes EMD-36856 and 8K3Q (apo state), EMD-36857 and 8K3R (GlcNAc incubated apo state), EMD-36863 and 8K3W (UDP-GlcNAc+GlcNAc bound state), EMD-36862 and 8K3V (UDP-GlcNAc bound state), EMD-36861 and 8K3U (primed UDP+GlcNAc bound state), EMD-36859 and 8K3T (UDP bound state), EMD-36855 and 8K3P (PolyB bound state), and EMD-36864 and 8K3X (NikkoZ bound state). Other models used for analysis in this paper can be accessed in RCSB PDB database with their respective accession codes: UDP-GlcNAc bound HAS (PDB ID: 7SP8), CesA (PDB ID: 7D5K), and *Candida albicans* Chs2 in apo (PDB ID: 7STL), UDP-GlcNAc bound (PDB ID: 7STM), NikkoZ bound (PDB ID: 7STN), and Polyoxin bound (PDB ID: 7STO) states. Source data are provided as a Source Data file. Source data are provided with this paper.

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

## Acknowledgements

Cryo-EM data were collected in the Cryo-EM Platform at the Center for Biological Imaging (CBI, cbi.ibp.ac.cn) at the Institute of Biophysics, Chinese Academy of Sciences, and Electron Microscopy Laboratory of Peking University. We thank Xuemei Li, Zhenxi Guo, Boling Zhu, Xujing Li, and Xiaojun Huang for facilitating data collection. We thank Kailong Li and Qing Li from Peking University for providing experimental instruments and materials. This work was supported by grants from the National Natural Science Foundation of China (32171212 to L.B., No. 32071207 to C.Y.), the Fundamental Research Funds for the Central Universities (to L.B.), Peking University (to L.B.), the National Basic Research Program of China (973 Program, No. 2012CB917202 to C.Y.).

## Author contributions

D.C., Z.W., C.Y., and L.B. conceived and designed the experiments. D.C., Z.W., L.W., P.Z., and L.B. performed the experiments. D.C., Z.W., C.Y., and

L.B. analyzed the data. L.B. wrote the manuscript with input from all authors.

## Competing interests

The authors declare no competing interests.
