## [Peer Review File · Nature Communications]

Structure, catalysis, chitin transport, and selective inhibition of chitin synthaseREVIEWER COMMENTS

Reviewer #1 (Remarks to the Author):

The manuscript by Dan-Dan Chen and colleagues describes several cryo EM structures and biochemical analyses of chitin synthase 1 from *S. cerevisiae*. The authors resolve several different states, including UDP bound, primed, and inhibited structures.

Chitin is one of the most abundant biopolymers on earth and of particular importance for fungal and arthropod development. Thus, insights into its biosynthesis are important.

Overall, the manuscript is well written and provides novel insights into CHS structure and function. However, I have several major concerns that should be addressed.

Major points:

My main concern relates to the map quality of the UDP-GlcNAc substrate and its interpretation. Considering the maps shown in Fig. S10, the authors hardly resolve any density corresponding to the donor sugar. In addition, as stated in the text, the donor sugar appears to be orientated away from its presumed binding pocket. I appreciate the authors' open discussion of the poor map quality, yet in light of the insufficiently resolved donor density, I don't think this state should be referred to as a substrate-bound state to avoid confusions in the future. Instead, this particular state could be interpreted as a 'partially inserted' substrate molecule, provided that convincing donor density is indeed observed. Alternatively, could it be that the observed states are post-hydrolysis states with an alternative UDP conformation?

Related to donor and acceptor binding, the authors do not mention the location and function of the base catalyst. Can this be included when discussing the primed or Nikkomycin inhibited states?

Map quality and modeling of the swapping loop: This study reveals a potential function for the swapping motif. The observation that deleting the swapping loop increases CHS' activity is interesting. However, this raises the question of how well the loop has been resolved in the various cryo EM maps. Based on Fig. 1d and S9, the loop appears to be fairly flexible. How confident can the authors be about the register of the swapping loop helix and the regions connecting it with the core structure?

Further, the authors state that the swapping loop helix is preceded by the conserved WGTKG motif but do not state what this motif is doing or where it is located. This seems to be an important feature of chitin synthases and should be discussed/described.

Considering the increased catalytic activity of the swapping loop deletion construct, I am wondering whether failure to complement the WT knockout is due to mis-localization of the deletion construct. Could it be that this construct is not delivered to the plasma membrane due to increased catalytic activity during trafficking?

Catalytic activity of CHS1: The authors use two assays to analyze in vitro catalytic activity. One quantifies the release of UDP and another utilizing wheat germ agglutinin to detect chitin oligosaccharides. First, the second assay should be described in more detail in the main text for non-experts. Second, could the authors provide an estimate of how long the in vitro synthesized chitin oligosaccharides are? Can the wheat germ agglutinin assay be calibrated, e.g. what is the minimal oligosaccharide length that is detected? This information is important to properly assess CHS1's catalytic activity.

Discussion of the transmembrane channel: The discussion of the TM channel and its gating is very difficult to follow. Particularly, Fig. 4a-c fails to illustrate the channel path and regions where it is closed to the extracellular or intracellular milieu. These panels should be revised.

Figure quality: Figure panels comparing different CHS1 conformations are generally very difficult to interpret. That is a particular concern for Fig. 3e, Fig. 4a-c, Fig. S13b, and Fig. 15. In addition, Fig. S13 is at very low resolution such that the sequences are hard to read.

Minor points:

Different detergents are used for catalytic assays and cryo EM structure determination. It should be stated in the methods why LMNG/CHS was not also used for the activity assays.

Page 7, line 189: VLPGA motif (residue 267-271). The numbers seem to be for a different protein.

Related to the above, line 234 states that the VLPGA motif was termed 'switch loop' based on this study, yet it seems that the authors follow a naming convention established for hyaluronan synthase. If so, this could be stated.

Page 10, line 260: Should Fig. 3e be 3d instead? Also, R718 and M454 are not shown in Fig. 3.

Page 12, last paragraph and Discussion: I am wondering whether Fig. 4 should be Fig. 5, and Fig. 5 should be Fig. 6?

Discussion, line 358: '...we propose that donor hydrolysis generates a disaccharide...' Why disaccharide? Shouldn't it be 'monosaccharide'?

Table 1: Please include a model-map correlation estimate as part of the refinement statistics.

Video 1: The video doesn't seem to illustrate major movements of TM4 or the swapping loop.

Reviewer #2 (Remarks to the Author):

The manuscript by Cai-Hong Yun, Lin Bai and coworkers describes structure determination of a *Saccharomyces cerevisiae* chitin synthase in the apo state or in the presence of substrates (GlcNAc, UDP-GlcNAc), product (UDP) or inhibitors (PolyD, NiZ). Chitin synthase has been long considered inappropriate for structure determination due to its zymogenic particularity, so that any information on structure characterization must be considered of high importance. Recently, a first cryo-EM structure determination of Chs from *Candida albicans* has been reported (reference 34 in the article), which opened new research perspectives. This initial work presented structures of caChs2 in the apo state or in complex with UDP-GlcNAc substrate or PolyD and NiZ inhibitors.

In the proposed manuscript, the cryo-EM determined structures of Chs1 in complex with inhibitors polyD and NiZ (lines 317-340) lead to the same conclusions than in reference 34 or in former studies dealing with CS inhibition. It is known for long that peptidyl-nucleosides act as UDP-GlcNAc competitors and structure-activity relationship studies allowed to determine the contribution of each part of the molecules and configurations to binding. The conclusions presented herein (lines 56-57, 63, 330, 340, 341-343) do not give an original lightning. A series of new discoveries are claimed by the authors, among which the « detailed catalytic cycle of scChs1 ». Although structures of scChs1 in prime state or in complex with GlcNAc+UDP-GlcNAc, or UDP-GlcNAc itself as well as GlcNAc or UDP as reaction products have been performed, no conclusion arises to explain the exact mechanism of action of chitin synthase. Some insights might be gained on a possible hydrolytic activity (lines 238-249) but the key O-C bond forming step between the donor and acceptor cannot be resolved by the given data. Rather, the observed positions of both putative substrates is not in agreement with a possible nucleophilic substitution at UDP-GlcNAc's C1' by O4 from GlcNAc (considered here as a initiator of the chitin chain) (lines 197-199). Other statements in the manuscript are in sharp contradiction with the general

understanding on chitin synthase : i) it is generally found that the purified protein is inactive ; here it was claimed that CS was functional after purification. The UDP-GlcNAc test (line 76-77) is questioning, since spontaneous hydrolysis of UDP-GlcNAc can not be ruled out (lines 253-254 !). Confirmation of such an unprecedented result should have been brought by a radioactive assay in addition to the WGA-coupled immuno HRP method. ii) The authors expressed and purified a truncated Chs1 (Chs1- Δ SL), lacking the swapping loop, in order to examine its role (151-162). Very surprisingly, the obtained protein, which appears as monomeric, is 10 times more active than the wild type according to enzymatic assays. Here again, such an astonishing result must be assessed by additional experiments, ie Mass spectra analysis to determine the monomeric state of the protein and a further radioactive enzymatic assay to verify the huge difference in activity. The conclusion of these experiments (« the swapping loop probably plays an inhibitory role in Chs1's activity ») appears as an over-simplification since many other hypotheses could be brought and verified experimentally, such as a conformational role. iii) complementation assays were conducted with Chs1- Δ SL to evaluate its capability to rescue Chs1 Δ phenotype (163-171). For these experiments my opinion is that the given data are not relevant (figure 2g) and can not support the conclusions. Cell images do not reflect the statements that « Chs1- Δ SL significantly disrupted Chs1 function in vivo as it was unable to rescue the Chs1 Δ yeast phenotype » (the 3rd and 4th images of figure 2g do not support this statement). And the final conclusion of this part is incorrect ; even if the experimental rescue results are correct with Chs1 Δ +Chs1 (positive rescue) and Chs1 Δ +Chs1- Δ SL (no rescue) one might not finish by saying that « the dimeric state of Chs1 is essential for its cellular function » (line 170-171). Indeed, Chs1- Δ SL is NOT an equivalent of monomeric Chs1.

In short, although structure determination of chitin synthases is a very attractive field of research, the results presented here do not meet the criteria for publication in Nature Com. Some conclusions arise from over-interpretation of experimental results or must be addressed by additional analyses. Finally, what would be the most attractive part of the paper, ie unraveling the mechanism of action of chitin synthases, affords no element explaining the essential and unknown part of the mechanism, ie how the attachment of donor and acceptor occurs. A two-active site mechanism has been proposed for chitin synthase in 2004, which is even not discussed or ruled out here.

Reviewer #3 (Remarks to the Author):

Overview of MS

This MS presents a structural and mechanistic study of the biosynthesis of chitin, one of the planet's most abundant polysaccharides, a topic of inherent interest to a broad audience. The authors present a cryogenic electron microscopic (cryo-EM) analysis of baker's yeast chitin synthase 1 (Chs1). The authors address the initiation of chitin chains, and demonstrate that Chs1 has hydrolytic activity towards its substrate, UDP-GlcNAc, and they present a mechanism in which the released GlcNAc then serves as a primer to initiate chitin chain synthesis. Further, it is shown that the synthase changes conformation depending on the presence of substrate (the sugar nucleotide UDP-GlcNAc), acceptor glycan (GlcNAc),

or of product (released UDP). Conformational changes cause a peptide loop to switch position upon binding of both UDP-GlcNAc and GlcNAc or UDP in the active site, opening a pore through which the nascent chitin chain can be translocated. It is further shown that Chs1 functions as a dimer, and structures are presented that show how nikkomycin and polyoxin, competitive inhibitors of chitin synthase, bind the enzyme.

General comments

The findings that Chs1 hydrolyzes UDP-GlcNAc, undergoes conformational changes that regulate access of growing product chain to translocation pore, and that Chs1 forms a dimer are significant contributions. However, the MS is undermined by two weaknesses. First, the effects of partial proteolytic treatment on enzyme activity are confusingly described and shed no light on proteolytic activation of the enzyme. Second, the finding of priming by free monomer (GlcNAc) was previously published, and the notion that this GlcNAc is generated by hydrolysis of the substrate UDP-GlcNAc was also proposed earlier. These issues are elaborated on below.

1) Effects of proteolytic treatment on Chs1 activity.

The phenomenon of proteolytic activation of certain chitin synthases, of which yeast Chs1 was the paradigm, was first reported half a century ago, but has still not been explained. The present MS, however, sheds no light on this phenomenon, despite the fact that the authors have powerful tools to do so. I raise the following concerns

i) The authors assert (lines 73-79) that affinity purified Chs1 has activity and it seems to be implied that this is without prior proteolytic treatment. However, no control experiments are presented to rule out the possibility that Chs1 might undergo proteolytic activation during purification. The authors assert that “apo Chs1” is in a lower active state, while protease-processed Chs1 is in a higher active state” (lines 87-89), but they do not show how the protease-processed form of Chs1 differs from the “apo-form “of Chs1. Is there any evidence from the cryo-EM studies for a proteolytic cleavage of Chs1? If there is no indication that the two forms of Chs1 differ, how can any increases in UDP-GlcNAc hydrolytic activity and chitin synthesis be explained? Because it is possible (maybe likely) that Chs1 became proteolytically activated during purification, it is imperative that the authors assay Chs1 activity of before solubilization and affinity purification with and without trypsin-pretreatment. If they detect large increases in chitin synthase activity in membrane fractions (as previously reported), then it would be reasonable to conclude that Chs1 had somehow become activated during purification, not that the Chs1 has high intrinsic activity.

ii) The authors' description of the effects of trypsin treatment is confusing and contains errors. Thus, the authors write (lines 79-86): "We found that purified Chs1 hydrolyzes UDP-Glc into free UDP with a glycosylation activity of ~33 nmol per mg protein per min and produces chitin with an activity of ~6.3 µg per mg protein per min. In contrast, the activity of Chs1 incubated with trypsin was ~93 nmol UDP per mg protein per min or ~10.0 µg chitin per mg protein per min, approximately 1.5-3 times that of the wild-type protein, which probably explains why the activity of Chs1 was found to be protease-activated in previous studies". Problem 1: shouldn't it be UDP-GlcNAc, not UDP-Glc? Problem 2: Using "glycosylation activity" in this context is misleading, because the UDP-Glo assay doesn't detect glycosyl transfer per se (rather, it involves a bioluminescence assay for ATP derived from UDP). Problem 3: What do the authors mean by "which probably explains why the activity of Chs1 was found to be protease-activated in previous studies"? This just restates the present, as well as many previous results, and is no explanation at all.

iii) The authors describe UDP-GlcNAc+GlcNAc-bound Chs1 as an "active state" (lines 176-177). Do they consider this state, and the proteolytically processed higher active state to be the same? If not, the two states should be distinguished.

iv) Have the authors obtained cryo-EM structures of the "apo" form before and after trypsin treatment? A clear demonstration of proteolysis leading to a conformation change of the enzyme to an active form would be a major contribution.

2) The account of priming of chitin synthases is inaccurate and the findings need to be presented in the context of prior published work.

i) In lines 239-240, the authors state that "Because of the lack of a free GlcNAc source in vivo, previous studies suggest that Chs1 performs "self-priming" by generating GlcNAc from UDP-GlcNAc", and they cite Refs 30 and 31 in support of this. This is inaccurate: neither of those cited papers (Cabib, 1987; Milewski et al., 2006) mentions the possibility that chitin synthases hydrolyze UDP-GlcNAc or use the released GlcNAc as primer. Quite the opposite: Cabib (1987) writes "Because... purified preparations of *S. cerevisiae* show very little stimulation by GlcNAc [citing Kang, M.S. et al., 1984; *J. Biol. Chem.* 259: 14966-14972] it is unlikely that the amino sugar functions as a primer": this can hardly be adduced as support for the notion of priming by GlcNAc!

ii) Importantly, other studies, not cited in this MS, have demonstrated priming experimentally and also discussed "self-priming". Work with related *S. cerevisiae* Chs2 showed that both free GlcNAc as well as 2-acetylamido glucosamine analogues prime formation of chitin oligosaccharides and insoluble chitin (Gyore, J., et al. (2014, *J. Biol. Chem.* 289: 12835-12841); Orlean, P. & Funai, D. (2018, *Cell Surf.* 5:100017)). Furthermore, the latter authors consider at length the possibility that Chs2 could hydrolyze UDP-GlcNAc and use the free GlcNAc so generated as primer. Not only that, they even used the term

“self-prime” for this. The present authors indeed provide compelling experimental support for these ideas, but it would be appropriate for them to cite the previous, highly relevant studies with yeast Chs2.

REVIEWER COMMENTS

Reviewer #1 (Remarks to the Author):

The manuscript by Dan-Dan Chen and colleagues describes several cryo EM structures and biochemical analyses of chitin synthase 1 from *S. cerevisiae*. The authors resolve several different states, including UDP bound, primed, and inhibited structures.

Chitin is one of the most abundant biopolymers on earth and of particular importance for fungal and arthropod development. Thus, insights into its biosynthesis are important. Overall, the manuscript is well written and provides novel insights into CHS structure and function. However, I have several major concerns that should be addressed.

Thanks for the comments. To answer the questions, we have performed new experiments and analysis. More details are provided as below.

Major points:

My main concern relates to the map quality of the UDP-GlcNAc substrate and its interpretation. Considering the maps shown in Fig. S10, the authors hardly resolve any density corresponding to the donor sugar. In addition, as stated in the text, the donor sugar appears to be orientated away from its presumed binding pocket. I appreciate the authors' open discussion of the poor map quality, yet in light of the insufficiently resolved donor density, I don't think this state should be referred to as a substrate-bound state to avoid confusions in the future. Instead, this particular state could be interpreted as a 'partially inserted' substrate molecule, provided that convincing donor density is indeed observed. Alternatively, could it be that the observed states are post-hydrolysis states with an alternative UDP conformation?

Thanks for the valuable suggestion. We agree with the reviewer that it's not accurate enough to simply name the map as UDP-GlcNAc bound state.

The UDP-GlcNAc bound state in the paper was generated by incubating Chs1 with UDP-GlcNAc for 10 min (UDP-GlcNAc-10min). The density of UDP group of UDP-GlcNAc in the map is very clear, while density of the GlcNAc group is relatively weaker. When we incubated Chs1 with UDP-GlcNAc for longer time (40 min), we captured the primed UDP+GlcNAc bound state (or named post-hydrolysis state), in which densities of UDP and GlcNAc clearly separate from each other. By comparing these two maps, both UDP and GlcNAc groups can be well assigned. Despite of weaker than UDP moiety, the density of GlcNAc moiety in UDP-GlcNAc bound state is very clear and the corresponding density is not available in primed state or UDP bound state (**Supplemental Figure 12**). Therefore, we think the primed map, other than UDP-GlcNAc-10min map, is likely to be in the post-hydrolysis state.

As the reviewer noticed, we found the GlcNAc moiety of UDP-GlcNAc orientated away from its presumed binding pocket in the UDP-GlcNAc bound state, and proposed the donor sugar “may need to rotate horizontally by 180° for the formation of a $\beta(1\rightarrow4)$ glycosidic bond”. In light of this, there are at least two UDP-GlcNAc bound states: 1) the current UDP-GlcNAc bound state as the UDP-GlcNAc loading state; 2) the predicted rotated UDP-GlcNAc as “inserted” state. It means the substrate is loaded onto the active site in our determined structure firstly (left panel of above figure), and then converts to an inserted position for following hydrolysis (an intermediate not detected in our work). After the substrate hydrolysis, the free GlcNAc will be inserted into the chitin transporting channel gate (right panel of above figure).

To be more accurate, we have revised corresponding text and changed the “UDP-GlcNAc bound” state to “UDP-GlcNAc loading” state.

Related to donor and acceptor binding, the authors do not mention the location and function of the base catalyst. Can this be included when discussing the primed or Nikkomycin inhibited states?

Thanks for the comment. We have added discussion in page9 on the base catalyst:”As the putative catalytic base residue, D602 also does not interact with UDP-GlcNAc in the structure. These findings suggest the GlcNAc moiety of UDP-GlcNAc may need to be rotated horizontally and inserted into active site for the formation of a $\beta(1\rightarrow4)$ glycosidic bond in the following reaction (**Fig. 3a**).”

Because we didn’t capture the “inserted state”, which can clearly show the function of D602 in donor sugar-phosphate bond cleavage, the discussion is mainly based on structures in other states.

Map quality and modeling of the swapping loop: This study reveals a potential function for the swapping motif. The observation that deleting the swapping loop increases CHS’ activity is interesting. However, this raises the question of how well the loop has been resolved in the various cryo EM maps. Based on Fig. 1d and S9, the loop appears to be fairly flexible. How confident can the authors be about the register of the swapping loop helix and the regions connecting it with the core structure?

Thanks for the comment. The densities of the swapping loop in our determined structures are relatively weaker than those in other parts of Chs1 or even invisible. To

better present its density, we prepared a new figure of the swapping loop in NikkoZ-bound state (**Supplementary Fig. 14a-b**). Although the density of swapping loop is relatively weaker, the map at a lower display threshold showed the swapping loop is connected to the core structure of Chs1. Especially for the swapping loop helix, the density displayed good sided chain features (**Supplementary Fig. 11**). Therefore, we are confident to assign the swapping loop to segment between T969 and T1024 of Chs1.

Besides, densities of the swapping loop in all seven different states in low threshold are shown as below.

Further, the authors state that the swapping loop helix is preceded by the conserved WGTKG motif but do not state what this motif is doing or where it is located. This seems to be an important feature of chitin synthases and should be discussed/described.

Thanks for the suggestion. In the revised manuscript, we highlighted the WGTKG motif in the map of Chs1 in **Supplementary Fig. 14a**, and also in the sequence in **Supplementary Fig.13**. Furthermore, we performed in vivo growth complementation assay of W969A and G970A in WGTKG motif (**Supplementary Fig. 14e, 15**). We found that both W969A and G970A were unable to rescue the *chs1* Δ yeast phenotype, indicating the important role of the WGTKG motif.

Considering the increased catalytic activity of the swapping loop deletion construct, I am wondering whether failure to complement the WT knockout is due to mis-localization of the deletion construct. Could it be that this construct is not delivered to the plasma membrane due to increased catalytic activity during trafficking?

Thanks for the question. We agree that it's possible failure to complement the WT knockout is due to mis-localization of the deletion construct. To answer this question, we tried to monitor the localization of Chs1- Δ SL by fusing GFP to C-terminal of Chs1- Δ SL. Unfortunately, the fluorescence signal was too weak to make a clear conclusion and thus was not included in the revised manuscript. The low signal is probably

because of extremely low expression level of Chs1- Δ SL (<0.01mg protein was purified from 10L cells). As shown in attached figure, it seems the expressed protein is partially localized in the plasma membrane, while many are also mis-localized (probably in ER membrane). In the revised manuscript, we added this possibility in page 8: "It's also possible that deletion of SL damages the localization of Chs1 in plasma membrane."

Catalytic activity of CHS1: The authors use two assays to analyze in vitro catalytic activity. One quantifies the release of UDP and another utilizing wheat germ agglutinin to detect chitin oligosaccharides. First, the second assay should be described in more detail in the main text for non-experts. Second, could the authors provide an estimate of how long the in vitro synthesized chitin oligosaccharides are? Can the wheat germ agglutinin assay be calibrated, e.g. what is the minimal oligosaccharide length that is detected? This information is important to properly assess CHS1's catalytic activity.

Thanks for the comments.

For the first question, we have added description of WGA in the main text "WGA is a dimeric lectin that exhibits high binding affinity for GlcNAc residue or oligomers", and also added more introduction of this assay in the method part "The assay mainly includes two steps: binding of synthesized chitin to a WGA-coated surface and detection of the polymer with a horseradish peroxidase-WGA conjugate. Horseradish peroxidase activity can be determined in absorbance at 600 nm, and the values are converted to amounts of chitin using commercial chitin as a standard."

For the second question, the minimal oligosaccharide detected by WGA-HRP assay is theoretically GlcNAc dimer, in which one sugar binds coated WGA and the other binds peroxidase-conjugated WGA. Previous study compared this WGA-HRP assay with conventional radioactive assay, and found their result and sensitivity are highly similar (as shown in below figure1). Besides, WGA-HRP binds longer-chain oligosaccharide with higher affinity (as shown in below figure2).

To further confirm in vitro catalytic activity of Chs1, we performed another chitin synthesis assay by calcofluor white (CFW) staining (**Supplementary Fig. 2**). CFW is a non-specific fluorochrome stain for 1,3- β and 1,4- β -linked polysaccharides and displays fluorescence when exposed to long wavelength ultraviolet. Accumulated chitin were detected in this in vitro chitin synthesis assay.

FIG. 4. Chitin synthase activities were assayed in a total membrane fraction from wild-type *S. cerevisiae*. The values for specific activities of Chs1, Chs2, and Chs3 determined by the chitin binding assay (black bars) are higher and less dispersed than the values obtained using the radioactive method (4) (gray bars). Error bars are SD for $n = 6$.

Figure 4 of Lucero, H.A., et al. *Analytical Biochemistry* 305, 97-105 (2002).

Competitive inhibition of WGA-HRP binding to GlcNAc as determined by ELLA

Compound	M_n^a (g mol ⁻¹)	Sugar		IC_{50} (nM)	r_p^c	r_p/n^d
		\bar{D}^a	motif n^b			
1			GlcNAc 1	$(17\,400 \pm 2800) \times 10^3$	1	1
26			(GlcNAc) ₂ 1	$(800 \pm 170) \times 10^3$	1	1
27			(GlcNAc) ₃ 1	$(107 \pm 39) \times 10^3$	1	1
28			(GlcNAc) ₄ 1	$70\,000 \pm 3000$	1	1
29			(GlcNAc) ₅ 1	$41\,000 \pm 9000$	1	1

Selected from Table 1 of Bojarová, P., et al. *Polymer Chemistry* 8(17): 2647-2658. (2017).

Moreover, according to other reviewer's comments, we confirmed that the activity of our old purified Chs1 was caused by partially proteolytic activation during purification. We performed many new experiments and structural analysis, including purification, SDS-PAGE, mass spectrometry, western-blot, etc. Our results showed activation of Chs1 zymogen by trypsin proteolysis is through removal of N-terminal region of Chs1, and the swapping loop is likely involved in the inhibitory role of N-terminal region by direct interaction. More details can be found in revised manuscript.

Discussion of the transmembrane channel: The discussion of the TM channel and its gating is very difficult to follow. Particularly, Fig. 4a-c fails to illustrate the channel path and regions where it is closed to the extracellular or intracellular milieu. These panels should be revised.

For better presentation, we have remade Fig. 4a-c, 4e, and also revised related text

accordingly. By changing the top view to side view, we think the new Fig.4e can better present key residues in the transmembrane channel.

These new figures clearly showed the cavity in the membrane, which features by a hydrophilic top part and a closed hydrophobic bottom. The hydrophilic environment of top part is more favorable to the chitin elongation and transport, while the bottom hydrophobic residues probably play gating function for chitin elongation and transport. Combining with following 3DVA analysis, our structures indicate the flexibility of TMH4 is correlated with the channel opening.

Figure quality: Figure panels comparing different CHS1 conformations are generally very difficult to interpret. That is a particular concern for Fig. 3e, Fig. 4a-c, Fig. S13b, and Fig. 15. In addition, Fig. S13 is at very low resolution such that the sequences are hard to read.

Thanks for the suggestion. We apologize for any inconvenience caused. We have remade Fig. 3e, Fig. 4a-c, Fig. S11 (S13 in revised paper), Fig. S13b (17b in revised paper) and Fig. S15 (S18 in revised paper). In general, for better presentation, we removed all unneeded elements, adjusted the view and colors. We also revised related text accordingly.

Minor points:

Different detergents are used for catalytic assays and cryo EM structure determination. It should be stated in the methods why LMNG/CHS was not also used for the activity assays.

Thanks for the suggestion. We have explained the reason for using DDM in the activity assays in the method part.

DDM is a popular mild non-ionic detergent, which has been proved to well suit for the solubilization, stabilization, and purification of membrane proteins. In this project, we used DDM/CHS to dissolve the membrane pellet, and then screened a range of detergents in the remaining step. We found LMNG/GDN/CHS is the best for preparing cryo-EM grid, but DDM/CHS is the best detergent for stability and yield of Chs1. To make the experiment simple, DDM/CHS purified sample was used in the activity assays.

Page 7, line 189: VLPGA motif (residue 267-271). The numbers seem to be for a different protein.

Related to the above, line 234 states that the VLPGA motif was termed 'switch loop' based on this study, yet it seems that the authors follow a naming convention established for hyaluronan synthase. If so, this could be stated.

We apologize for the mistake. The residue number was fixed.

We also revised related text of switch loop and cited the paper of hyaluronan synthase.

Page 10, line 260: Should Fig. 3e be 3d instead? Also, R718 and M454 are not shown in Fig. 3.

Fig. 3e was revised to Fig. 3d.

R718 and M454 were added in Fig. 3a and 3e.

Page 12, last paragraph and Discussion: I am wondering whether Fig. 4 should be Fig. 5, and Fig. 5 should be Fig. 6?

Revised.

Discussion, line 358: ‘...we propose that donor hydrolysis generates a disaccharide...’ Why disaccharide? Shouldn’t it be ‘monosaccharide’?

Thanks for the comment. We revised the sentence to “We further propose that the following hydrolysis of the second donor molecule generates a free GlcNAc molecule, which then forms a disaccharide with the acceptor GlcNAc via β (1→4) glycosidic bond.”

Table 1: Please include a model-map correlation estimate as part of the refinement statistics.

We have added the FSC(model-map) at a correlation cutoff value of 0.5 in the statistics table. The FSC(model-map) were generated by real-space refinement in the PHENIX program.

Video 1: The video doesn’t seem to illustrate major movements of TM4 or the swapping loop.

Thanks for the comment. As the reviewer pointed out, the 3DVA analysis only revealed the flexibility of the domain swapping loop and TMH4. Such flexibility indicated the domain swapping loop and TMH4 are movable, but didn’t illustrate the movement path. To be more accurate, we have revised the text accordingly, and avoid to use the “movement”.

Reviewer #2 (Remarks to the Author):

The manuscript by Cai-Hong Yun, Lin Bai and coworkers describes structure determination of a *Saccharomyces cerevisiae* chitin synthase in the apo state or in the presence of substrates (GlcNAc, UDP-GlcNAc), product (UDP) or inhibitors (PolyD, NiZ). Chitin synthase has been long considered inappropriate for structure determination due to its zymogenic particularity, so that any information on structure characterization must be considered of high importance. Recently, a first cryo-EM structure determination of Chs from *Candida albicans* has been reported (reference 34 in the article), which opened new research perspectives. This initial work presented

structures of caChs2 in the apo state or in complex with UDP-GlcNAc substrate or PolyD and NiZ inhibitors.

In the proposed manuscript, the cryo-EM determined structures of Chs1 in complex with inhibitors polyD and NiZ (lines 317-340) lead to the same conclusions than in reference 34 or in former studies dealing with CS inhibition. It is known for long that peptidyl-nucleosides act as UDP-GlcNAc competitors and structure-activity relationship studies allowed to determine the contribution of each part of the molecules and configurations to binding. The conclusions presented herein (lines 56-57, 63, 330, 340, 341-343) do not give an original lightning.

Thanks for the comments. We have revised the text for better presentation, and highlighted findings of the peptidyl moiety of inhibitor.

As the reviewer pointed out, it is known for long that peptidyl-nucleosides act as UDP-GlcNAc competitors and functional studies also provided some insights into the contribution of each part of the molecules. However, determining the structures of chitin synthase in complex with these inhibitors can reveal the inhibitory mechanism in atomic level, including atom-atom interactions. For example, while the nucleoside moiety of inhibitor is likely competing to UDP of UDP-GlcNAc substrate, it's unclear how the peptidyl moiety of inhibitors binds Chs1. The structural studies can answer these questions and are also very helpful for following structure based drug improvement and drug development.

In revised section of "Mechanism of peptidyl nucleoside inhibitors of chitin synthase", we highlighted the unique architecture of PolyD and NikkoZ in our structures. Notably, the structures provided novel findings that the peptidyl moiety of peptidyl nucleoside inhibitors induces the switch loop open and extend toward the chitin transport channel. The end part of the peptidyl moieties of PolyB and NikkoZ occupy the GlcNAc acceptor binding site, and thus block the chitin transport channel. Therefore, our structures suggest that the peptidyl nucleoside inhibitors inhibit chitin synthases by using their nucleoside moiety competing with UDP binding, and their peptidyl moiety inducing the switch loop open and blocking the gate of chitin transport channel.

A series of new discoveries are claimed by the authors, among which is the « detailed catalytic cycle of scChs1 ». Although structures of scChs1 in prime state or in complex with GlcNAc+UDP-GlcNAc, or UDP-GlcNAc itself as well as GlcNAc or UDP as reaction products have been performed, no conclusion arises to explain the exact mechanism of action of chitin synthase. Some insights might be gained on a possible hydrolytic activity (lines 238-249) but the key O-C bond forming step between the donor and acceptor cannot be resolved by the given data. Rather, the observed positions of both putative substrates is not in agreement with a possible nucleophilic substitution at UDP-GlcNAc's C1' by O4 from GlcNAc (considered here as a initiator of the chitin chain) (lines 197-199).

Thanks for the comments.

As an inverting glycosyltransferase, chitin synthases utilize a direct displacement SN₂-like mechanism, in which the chitin synthase catalyzes the nucleophilic attack of oxygen on the anomeric carbon with simultaneous cleavage of the bond to the phosphate-containing leaving group. We totally agree with reviewer that the O-C bond forming step (GlcNAc's O4 and UDP-GlcNAc's C1') is absolutely a key step in the catalytic cycle. However, to clearly show details of donor sugar-phosphate bond cleavage and new O-C bond forming, a "substrate inserted state" (in which the catalytic base directly contacts with the O-C bond) is probably needed. We tried to capture this state during the revision, but unfortunately failed. Therefore, we revised the text and discussed the function of putative catalytic base residue basing on our structures: "the GlcNAc moiety of UDP-GlcNAc is located right above IF2 in a bent configuration extending outward from the active siteAs the putative catalytic base residue, D602 does not interact with UDP-GlcNAc in the structure. These findings suggest the GlcNAc moiety of UDP-GlcNAc may need to rotate horizontally by 180° for the formation of a β (1→4) glycosidic bond in the following reaction(Fig. 3a)." We hope we can completely reveal the O-C bond forming step in future.

Besides, our structures of Chs1 in different states (donor bound, donor+acceptor bound, and products bound) have provided extensive insights into other steps of the catalytic cycle of Chs1, including the initial binding of UDP-GlcNAc, proposed rotation of GlcNAc moiety of UDP-GlcNAc, movement of GTD during UDP-GlcNAc hydrolysis, requirement and participation of Mg²⁺, position of hydrolyzed products. These findings are also very helpful for us to fully understand the catalytic mechanism of Chs1.

Other statements in the manuscript are in sharp contradiction with the general understanding on chitin synthase :

i) it is generally found that the purified protein is inactive ; here it was claimed that CS was functional after purification. The UDP-Glo™ test (line 76-77) is questioning, since spontaneous hydrolysis of UDP-GlcNAc can not be ruled out (lines 253-254 !). Confirmation of such an unprecedented result should have been brought by a radioactive assay in addition to the WGA-coupled immuno HRP method.

Thanks for the comments. Firstly, we apologize for the inaccurate description in lines 253-254 (line 308 in revised manuscript) and misunderstanding caused. This sentence doesn't mean "hydrolysis of UDP-GlcNAc" and was revised to "which indicates small part of Chs1 molecules in this sample have been self-primed". In our UDP-Glo assays, buffer containing UDP-GlcNAc but without Chs1 was used as the control (**Fig.1b**), and only showed very low signal. Therefore, it won't affect final conclusion of the UDP-Glo assays.

To further confirm whether our purified sample is active, we performed a new chitin synthesis assay by staining the product using calcofluor white (CFW) (**Supplemental Figure 2**, attached below), which clearly proved our purified Chs1 is active in direct observation. We also apologize for not performing radioactive assay as the reviewer suggested because of limitation of experimental conditions (no allowed lab for

radioactive assay in our school).

As the reviewer pointed out, our result is in contrary with previous works that Chs1 is inactive *in vitro* and could be activated by proteolysis. To answer this question, we first performed experiment to verify whether our purified sample has already been partially digested during purification or not.

We performed new purification more carefully by adding excess protease inhibitors in all buffers, keeping sample always at 4°C, and shorten the purification time. Indeed, the peak of Chs1 in gel filtration (green) was shifted left slightly compare to the old peak (blue) (**Supplemental Figure 1**, panel A as attached). Furthermore, while SDS-PAGE of old purified sample (blue frac14-15ml) only contains a single band lower than 130kDa marker (panel B), SDS-PAGE of the new purified sample clearly showed two bands (higher and lower than 130kDa marker respectively, panel C). To figure out the identities of these two bands, we performed anti-flag western-blot assay of cell membrane (following previous studies) and purified Chs1. We found both of the two proteins of purified sample belong to Chs1, while the higher band is the same as Chs1 in membrane and likely corresponding to intact Chs1 (panel D). We further performed tryptic digestion mass spectrometry for the two bands in panel C (highlighted by red and yellow stars). The results supported that the higher band is intact Chs1, and indicated the lower band lacks N-terminal region of Chs1 (panel E and panel F). It's also consistent with our cryo-EM structures, in which most part of Chs1 was well resolved except the N-terminal region.

Taking together, these results indicated our purified Chs1 has already been partially degraded in N-terminal region by endogeneous protease, and thus was partially activated. Although we tried hardly to do purification with more care, purified sample still contained the cleaved band, suggesting the N-terminal region is easily to be degraded.

Furthermore, to understand how trypsin proteolysis activates Chs1, we incubated purified Chs1 with trypsin in different mass ratios and times, and analyzed the product by SDS-PAGE and tryptic digestion mass spectrometry (**Supplementary Fig. 3a-c**). We showed proteolysis of purified Chs1 by trypsin generated a product with molecular weight at ~35-40kDa less than wild type Chs1 (panel A-B), and the first ~340 residues in N-terminus of Chs1 is likely removed by trypsin treatment (panel C). To further confirm this finding, we expressed and purified a truncated Chs1 by deleting N-terminal 340 residues (Chs1- Δ N) (panel A). We found Chs1- Δ N is about in the same size as the digested product of purified Chs1 by trypsin, and was not sensitive to trypsin proteolysis as earlier purified sample. The finding is also

consistent with our cryo-EM structure of Chs1, in which first ~380 residues of Chs1 are largely disordered and thus more sensitive to trypsin proteolysis (Fig. 1d-g). These findings revealed that proteolysis activation of Chs1 zymogen is to remove the N-terminal region (NTR) of Chs1. It also indicated the N-terminal region of Chs1 zymogen likely plays an inhibitory function.

ii) The authors expressed and purified a truncated Chs1 (Chs1-ΔSL), lacking the swapping loop, in order to examine its role (151-162).

Very surprisingly, the obtained protein, which appears as monomeric, is 10 times more active than the wild type according to enzymatic assays. Here again, such an astonishing result must be assessed by additional experiments, ie Mass spectra analysis to determine the monomeric state of the protein and a further radioactive enzymatic assay to verify the huge difference in activity. The conclusion of these experiments (« the swapping loop probably plays an inhibitory role in Chs1's activity ») appears as an over-simplification since many other hypotheses could be brought and verified experimentally, such as a conformational role.

Thanks for the comments. Firstly, we performed a new chitin synthesis assay by staining the product using calcofluor white (CFW). Consistent with the UDP-Glo assay and WGA-coupled HRP assay, this new data also supported Chs1-ΔSL is much more active (Supplementary Fig. 2).

To further determine the state of Chs1-ΔSL, we performed blue-native PAGE gel (Supplementary Fig. 14c). The result showed most Chs1-ΔSL is in monomeric state,

and was in agreement with its gel filtration elution volume.

We also performed SDS-PAGE gel of purified Chs1- Δ SL, and found its N-terminal region has been largely degraded during purification as processed by trypsin (**Supplementary Fig. 14d**). As mentioned above, activation of Chs1 zymogen relies on the proteolytic release of N-terminal loop. Accordingly, we revised the conclusion that « the swapping loop probably plays an inhibitory role in Chs1's activity » to “These results showed removal of swapping loop make the N-terminal inhibitory region of Chs1 more sensitive to proteolysis and thus enhance its activity significantly.”

Interestingly, our cryo-EM map of Chs1 showed some extra unassigned density locating next to the swapping loop likely belong to the NTR of Chs1 (**Supplementary Fig. 16a-c**). It indicated the NTR is partly stabilized by its interactions with the swapping loop and GTD. Conceivably, removal of SL of Chs1 helps to release the N-terminal region and thus activate Chs1's activity. This finding can explain why the N-terminal region of Chs1- Δ SL has been fully digested during purification and the dramatically increased activity. The swapping loop is likely involved in the inhibitory role of NTR of Chs1 by direct interaction.

iii) complementation assays were conducted with Chs1- Δ SL to evaluate its capability to rescue Chs1 Δ phenotype (163-171). For these experiments my opinion is that the given data are not relevant (figure 2g) and can not support the conclusions. Cell images do not reflect the statements that « Chs1- Δ SL significantly disrupted Chs1 function in vivo as it was unable to rescue the Chs1 Δ yeast phenotype » (the 3rd and 4th images of figure 2g do not support this statement). And the final conclusion of this part is incorrect; even if the experimental rescue results are correct with Chs1 Δ +Chs1 (positive rescue) and Chs1 Δ +Chs1- Δ SL (no rescue) one might not finish by saying that « the dimeric state of Chs1 is essential for its cellular function » (line 170-171). Indeed, Chs1- Δ SL is NOT an equivalent of monomeric Chs1.

Thanks for the comments. We agree with the reviewer that the descriptions are not accurate. We carefully analyzed the data and showed number of cell strings of Chs1- Δ SL is about 60% of chs1 Δ yeast in statistic (**Fig. 2g, Supplementary Fig.14e, 15**). We also revised the demonstration of Fig. 2g from “was unable to rescue the chs1 Δ yeast phenotype” to “Chs1- Δ SL was unable to fully rescue the chs1 Δ yeast phenotype like the wild-type Chs1”.

Besides, we further explained that the rescue experiment was performed because “As monomeric Chs1- Δ SL was highly active in vitro, we wonder whether Chs1- Δ SL can replace the function of dimeric Chs1 in vivo or not.” We removed earlier conclusion, and revised to “In contrast, Chs1- Δ SL was unable to fully rescue the chs1 Δ yeast phenotype like the wild-type Chs1, indicating the swapping loop is essential for regular cellular function of Chs1. Disrupted dimeric state or excessive activity of Chs1- Δ SL are possible reasons for this. It’s also possible that deletion of SL damages the localiztion of Chs1 in plasma membrane.”

In short, although structure determination of chitin synthases is a very attractive field of research, the results presented here do not meat the criteria for publication in Nature Com. Some conclusions arise from over-interpretation of experimental results or must be addressed by additional analyses. Finaly, what would be the most attractive part of

the paper, ie unraveling the mechanism of action of chitin synthases, affords no element explaining the essential and unknown part of the mechanism, ie how the attachment of donor and acceptor occurs. A two-active site mechanism has been proposed for chitin synthase in 2004, which is even not discussed or ruled out here.

Thanks for the reviewer's valuable comments. As mentioned above, we have performed new experiments and revised the texts accordingly. Notably, our new data explained how trypsin proteolysis activates Chs1 zymogen (by removal of N-terminal region of Chs1). Although our results couldn't explain the O-C bond forming step (a substrate inserted state is needed for this question), the structures of Chs1 in different states have provided extensive insights into other steps of the catalytic cycle of Chs1, including the initial binding of UDP-GlcNAc, proposed rotation of GlcNAc moiety of UDP-GlcNAc, movement of GTD during UDP-GlcNAc hydrolysis, requirement and participation of Mg^{2+} , position of hydrolyzed products. These findings are also very helpful for us to fully understand the catalytic mechanism of Chs1.

Besides, we cited and discussed the reference mentioned by the reviewer. Different from the two-active sites model, our structures showed each subunit only has one active site, and two active sites in Chs1 dimer are too far to cooperate with each other. Further research is still required to fully understand the mechanism of chitin elongation of Chs1, including how to connect GlcNAc residues in alternating orientations.

Reviewer #3 (Remarks to the Author):

Overview of MS

This MS presents a structural and mechanistic study of the biosynthesis of chitin, one of the planet's most abundant polysaccharides, a topic of inherent interest to a broad audience. The authors present a cryogenic electron microscopic (cryo-EM) analysis of baker's yeast chitin synthase 1 (Chs1). The authors address the initiation of chitin chains, and demonstrate that Chs1 has hydrolytic activity towards its substrate, UDP-GlcNAc, and they present a mechanism in which the released GlcNAc then serves as a primer to initiate chitin chain synthesis. Further, it is shown that the synthase changes conformation depending on the presence of substrate (the sugar nucleotide UDP-GlcNAc), acceptor glycan (GlcNAc), or of product (released UDP). Conformational changes cause a peptide loop to switch position upon binding of both UDP-GlcNAc and GlcNAc or UDP in the active site, opening a pore through which the nascent chitin chain can be translocated. It is further shown that Chs1 functions as a dimer, and structures are presented that show how nikkomycin and polyoxin, competitive inhibitors of chitin synthase, bind the enzyme.

General comments

The findings that Chs1 hydrolyzes UDP-GlcNAc, undergoes conformational changes

that regulate access of growing product chain to translocation pore, and that Chs1 forms a dimer are significant contributions. However, the MS is undermined by two weaknesses. First, the effects of partial proteolytic treatment on enzyme activity are confusingly described and shed no light on proteolytic activation of the enzyme. Second, the finding of priming by free monomer (GlcNAc) was previously published, and the notion that this GlcNAc is generated by hydrolysis of the substrate UDP-GlcNAc was also proposed earlier. These issues are elaborated on below.

Thanks for the reviewer's valuable comments. To answer these questions, we have performed many new experiments and revised the texts accordingly. Notably, our new data showed activation of Chs1 zymogen by trypsin proteolysis is through removal of N-terminal region of Chs1.

1) Effects of proteolytic treatment on Chs1 activity.

The phenomenon of proteolytic activation of certain chitin synthases, of which yeast Chs1 was the paradigm, was first reported half a century ago, but has still not been explained. The present MS, however, sheds no light on this phenomenon, despite the fact that the authors have powerful tools to do so. I raise the following concerns

i) The authors assert (lines 73-79) that affinity purified Chs1 has activity and it seems to be implied that this is without prior proteolytic treatment. However, no control experiments are presented to rule out the possibility that Chs1 might undergo proteolytic activation during purification. The authors assert that "apo Chs1" is in a lower active state, while protease-processed Chs1 is in a higher active state" (lines 87-89), but they do not show how the protease-processed form of Chs1 differs from the "apo-form" of Chs1. Is there any evidence from the cryo-EM studies for a proteolytic cleavage of Chs1? If there is no indication that the two forms of Chs1 differ, how can any increases in UDP-GlcNAc hydrolytic activity and chitin synthesis be explained? Because it is possible (maybe likely) that Chs1 became proteolytically activated during purification, it is imperative that the authors assay Chs1 activity of before solubilization and affinity purification with and without trypsin-pretreatment. If they detect large increases in chitin synthase activity in membrane fractions (as previously reported), then it would be reasonable to conclude that Chs1 had somehow become activated during purification, not that the Chs1 has high intrinsic activity.

As the reviewer pointed out, the activity of our purified Chs1 may be caused by proteolytic activation during purification. Therefore, we further performed new purification, and analyzed our purified sample using SDS-PAGE, mass spectrometry and western-blot. Details are shown as below.

We performed new purification more carefully by adding excess protease inhibitors in all buffers, keeping sample always at 4°C, and shorten the purification time. Indeed, the peak of Chs1 in gel filtration (green) was shifted left slightly compare to the old peak (blue) (**Supplemental Figure 1**, panel A as attached below). Furthermore, while

SDS-PAGE of old purified sample (blue frac14-15ml) only contains a single band lower than 130kDa marker (panel B), SDS-PAGE of the new purified sample clearly showed two bands (higher and lower than 130kDa marker respectively, panel C). To figure out the identities of these two bands, we performed anti-flag western-blot assay of cell membrane (following previous studies) and purified Chs1. We found both of the two proteins of purified sample belong to Chs1, while the higher band is the same as Chs1 in membrane and likely corresponding to intact Chs1 (panel D). We further performed tryptic digestion mass spectrometry for the two bands in panel C (highlighted by red and yellow stars). The results supported that the higher band is intact Chs1, and indicated the lower band lacks N-terminal region of Chs1 (panel E and panel F). It's also consistent with our cryo-EM structures, in which most part of Chs1 was well resolved except the N-terminal region.

Taking together, these results indicated our purified Chs1 has already been partially degraded in N-terminal region by endogeneous protease, and thus was partially activated. Although we tried hardly to do purification with more care, purified sample still contained the cleaved band, suggesting the N-terminal region is easily to be degraded.

Furthermore, to understand how trypsin proteolysis activates Chs1, we incubated purified Chs1 with trypsin in different mass ratios and times, and analyzed the product by SDS-PAGE and tryptic digestion mass spectrometry (**Supplementary Fig. 3a-c**, as attached below). We showed proteolysis of purified Chs1 by trypsin generated a product with molecular weight at ~35-40kDa less than wild type Chs1 (panel A-B), and the first ~340 residues in N-terminus of Chs1 is likely removed by trypsin treatment (panel C). To further confirm this finding, we expressed and purified a truncated Chs1 by deleting N-terminal 340 residues (Chs1-ΔN) (panel A). We found Chs1-ΔN is about in the same size as the digested product of purified Chs1 by trypsin, and was not sensitive to trypsin proteolysis as earlier purified sample. The finding is also consistent with our cryo-EM structure of Chs1, in which first ~380 residues of Chs1 are largely disordered and thus more sensitive to trypsin proteolysis (**Fig. 1d-g**). These findings revealed that proteolysis activation of Chs1 zymogen is to remove the N-terminal region (NTR) of Chs1. It also indicated the N-terminal region of Chs1 zymogen likely plays an inhibitory function.

We also performed SDS-PAGE gel of purified Chs1- Δ SL, and found its N-terminal region has been largely degraded during purification as processed by trypsin (**Supplementary Fig. 14d**, as attached below). As mentioned above, activation of Chs1 zymogen relies on the proteolytic release of N-terminal loop. Accordingly, we revised the conclusion that « the swapping loop probably plays an inhibitory role in Chs1's activity » to "These results showed removal of swapping loop make the N-terminal inhibitory region of Chs1 more sensitive to proteolysis and thus enhance its activity significantly."

By analyzing our cryo-EM structures, we also tried to explain why the N-terminal region of Chs1- Δ SL was easily digested during purification and its activity was dramatically increased. Our cryo-EM map of Chs1 showed some extra unassigned density locating next to the swapping loop likely belong to the NTR of Chs1 (**Supplementary Fig. 16a-c**, as attached below). It indicated the NTR is partly stabilized by its interactions with the swapping loop and GTD. Conceivably, removal of SL of Chs1 helps to release the N-terminal region and thus activate Chs1's activity. The swapping loop is likely involved in the inhibitory role of NTR of Chs1 through direct interaction.

Besides, according to reviewer's suggestion, we also performed cryo-EM analysis for the Chs1 treated with trypsin to see whether trypsin proteolysis alter the structure of Chs1. But unfortunately we didn't get high resolution structure using trypsin treated sample because the cryo-EM grid preparation parameter changed and need more time to be optimized. As this manuscript already contains a large amount of work and novel insights, we plan to perform it in the future.

ii) The authors' description of the effects of trypsin treatment is confusing and contains errors. Thus, the authors write (lines 79-86): "We found that purified Chs1 hydrolyzes UDP-Glc into free UDP with a glycosylation activity of ~33 nmol per mg protein per min and produces chitin with an activity of ~6.3 μ g per mg protein per min. In contrast, the activity of Chs1 incubated with trypsin was ~93 nmol UDP per mg protein per min or ~10.0 μ g chitin per mg protein per min, approximately 1.5-3 times that of the wild-type protein, which probably explains why the activity of Chs1 was found to be protease-activated in previous studies". Problem 1: shouldn't it be UDP-GlcNAc, not UDP-Glc?

The typo was fixed.

Problem 2: Using "glycosylation activity" in this context is misleading, because the UDP-Glo assay doesn't detect glycosyl transfer per se (rather, it involves a bioluminescence assay for ATP derived from UDP).

This paragraph had been rewritten and the "glycosylation activity" was not used.

Problem 3: What do the authors mean by "which probably explains why the activity of Chs1 was found to be protease-activated in previous studies"? This just restates the present, as well as many previous results, and is no explanation at all.

This paragraph had been rewritten. With new data, we showed proteolysis activation of Chs1 zymogen is by removal of the N-terminal region of Chs1.

iii) The authors describe UDP-GlcNAc+GlcNAc-bound Chs1 as an “active state” (lines 176-177). Do they consider this state, and the proteolytically processed higher active state to be the same? If not, the two states should be distinguished.

As mentioned above, the sample we used for cryo-EM analysis was partially digested for the first ~160 residues, and the inhibitory N-terminal region of Chs1 zymogen digested by trypsin is ~340 residues. Therefore, our structure is not totally equal to the proteolytically processed sample in the N-terminal region. However, densities of first ~380 residues of Chs1 in our cryo-EM maps were very weak or even invisible (only little weak densities locating on top of GTD), it means removal of N-terminal region by trypsin won't affect the overall structure of Chs1, especially for the substrate binding sites. Because UDP-GlcNAc+GlcNAc-bound Chs1 was mainly described to demonstrate the catalytic mechanism, the two states should be the same in the active sites. To be more concise, we think it's better to not distinguish them in this manuscript.

iv) Have the authors obtained cryo-EM structures of the “apo” form before and after trypsin treatment? A clear demonstration of proteolysis leading to a conformation change of the enzyme to an active form would be a major contribution.

According to reviewer's suggestion, we performed cryo-EM analysis for the Chs1 treated with trypsin to see whether trypsin proteolysis alter the structure of Chs1. But unfortunately we didn't get high resolution structure using trypsin treated sample because the cryo-EM grid preparation parameter changed and need more time to be optimized. As this MS already contains a large amount of work and novel insights, we plan to perform it in the future.

Our new data showed that the trypsin is likely cleave the first ~340 residues of Chs1. Densities of first ~380 residues of Chs1 in our cryo-EM maps were very weak or even invisible (only little weak densities locating on top of GTD), it means removal of N-terminal region by trypsin won't affect the overall structure of Chs1. Basing on the following study of swapping loop and structures, we propose that the inhibitory N-terminal region may inhibit the function of Chs1 by limiting the movement of domain swapping loop and GTD. Of course, a high resolution structure of trypsin treated Chs1 would confirm this proposition. We hope we can determine this structure in the future.

2) The account of priming of chitin synthases is inaccurate and the findings need to be presented in the context of prior published work.

i) In lines 239-240, the authors state that “Because of the lack of a free GlcNAc source in vivo, previous studies suggest that Chs1 performs “self-priming” by generating GlcNAc from UDP-GlcNAc”, and they cite Refs 30 and 31 in support of this. This is inaccurate: neither of those cited papers (Cabib, 1987; Milewski et al., 2006) mentions the possibility that chitin synthases hydrolyze UDP-GlcNAc or use the released GlcNAc as primer. Quite the opposite: Cabib (1987) writes “Because... purified preparations of *S. cerevisiae* show very little stimulation by GlcNAc [citing Kang, M.S. et al., 1984; J.

Biol. Chem. 259: 14966-14972] it is unlikely that the amino sugar functions as a primer": this can hardly be adduced as support for the notion of priming by GlcNAc!

We apologize for not citing the reference correctly and have revised the text following reviewer's next comment to:

"Initiation of chitin synthesis requires GlcNAc acceptor binding in the active site of Chs1. Revealing how to generate the first GlcNAc is important to understand the mechanism of chitin synthase. Because of lacking a free GlcNAc source in vivo, previous studies suggest that chitin synthases can "self-prime" by generating GlcNAc from UDP-GlcNAc with an intrinsic UDP-GlcNAc hydrolyzing activity (cite Gyore, J., et al. 2014, and Orlean, P. & Funai, D. 2018)".

ii) Importantly, other studies, not cited in this MS, have demonstrated priming experimentally and also discussed "self-priming". Work with related *S. cerevisiae* Chs2 showed that both free GlcNAc as well as 2-acylamido glucosamine analogues prime formation of chitin oligosaccharides and insoluble chitin (Gyore, J., et al. (2014, *J. Biol. Chem.* 289: 12835-12841); Orlean, P. & Funai, D. (2018, *Cell Surf.* 5:100017)). Furthermore, the latter authors consider at length the possibility that Chs2 could hydrolyze UDP-GlcNAc and use the free GlcNAc so generated as primer. Not only that, they even used the term "self-prime" for this. The present authors indeed provide compelling experimental support for these ideas, but it would be appropriate for them to cite the previous, highly relevant studies with yeast Chs2.

Thanks for the reviewer's valuable comments. Indeed, our results are well consistent with these findings. We have revised the text and cited the references mentioned by the reviewer.

REVIEWER COMMENTS

Reviewer #1 (Remarks to the Author):

The revised manuscript by Dan-Dan Chen et al. has improved slightly, yet several important concerns have not been addressed.

Main concerns:

1. Chs1 activation: Prompted by the reviewers, the authors attempted to address the activation mechanism of Chs1. The observation that a swapping loop deleted Chs1 version exhibits significantly increased catalytic activity compared to the wild-type enzyme is intriguing. Now, the authors argue that swapping loop deletion renders the enzyme's N-terminal domain more prone to proteolysis and that its removal may account for activation. This idea is supported by the observed proteolysis of the wild-type enzyme and swapping loop mutant during purification.

To address this point, the authors generated an N-terminally truncated Chs1 construct equivalent to the remaining proteolytic fragment and show that this construct resists trypsin digestion. Accordingly, the authors conclude that the N-terminal domain is autoinhibitory and that its removal activates the enzyme. However, the critical experiment showing that the N-terminally truncated Chs1 version is catalytically active (without trypsin digestion) comparable to the swapping loop mutant or a trypsin digested wild-type enzyme is not shown. Hence, the conclusion that removal of the N-terminus activates Chs1 is not supported.

It is also worth noting that the observed increase in activity of the trypsin digested and swapping loop deleted constructs (relative to full length) is not equivalent. Trypsin digestion appears to increase the wild-type activity about 3-fold (Fig. 1b), whereas swapping loop removal accounts for an increase of about 200-fold (Fig. 2e)

Further, the WGA chitin synthesis assays would benefit from a chitinase control reaction where chitin is specifically degraded prior to WGA binding.

2. Substrate-bound states: In my opinion, the EM maps presented for UDP-GlcNAc in the absence and the presence of a priming monosaccharide are insufficient to be interpreted as UDP-GlcNAc molecule. Fig. S12 hardly shows any density for the donor sugar, perhaps with the exception of its acetamido group (although even this is not clear). This should not be interpreted as a donor-bound state and also not as a substrate-loaded state that needs to rearrange for glycosyl transfer. Accordingly, the corresponding discussion of different binding poses of UDP versus UDP-GlcNAc ligands can either be removed or modified significantly, considering the uncertainty related to the presumed substrate-bound states.

3. Acceptor positioning: The authors modeled a GlcNAc monosaccharide at the acceptor binding site but do not discuss how this ligand is coordinated. Is its acetamido group resolved to allow proper positioning?

4. Base catalyst: The authors refer to Asp602 as the base catalyst (line 244), yet according to the sequence alignment (Fig. S13), this residue belongs to a DAG motif presumably involved in cation/nucleotide binding. Could it be that Asp717 is the base catalyst?

5. Swapping loop modeling: I previously requested clarification on how the swapping loop was modeled, considering the poor map quality presented for this region. Based on the information presented in the rebuttal letter and the supplement, I cannot imagine how this loop could have been modeled to allow discussing individual side chain locations (lines 171-173). Perhaps a backbone model would suffice (or an AlphaFold model docked into the map, but this should be clearly stated)?

6. Discussion, line 420-423: "...the following hydrolysis of the second donor molecule generates a free GlcNAc molecule, which then forms a disaccharide with the acceptor....". This sentence is almost certainly incorrect. GT-2 enzymes are believed to utilize an inverting SN₂-like nucleophilic displacement mechanism, which means that the acceptor hydroxyl (C4 of the GlcNAc monosaccharide) attacks the C1 carbon of the donor sugar, such that its configuration is inverted from alpha to beta. Thus, the substrate molecule is not first hydrolyzed and then attached to the primer as this reaction requires an activated C1 carbon.

Minor points:

Fig. 1b: It would be informative to test whether trypsin activated Chs1 is still sensitive to NikkoZ or PolyB inhibition.

Fig. 4b and c: These panels are very difficult to interpret, certainly as a printed version. Perhaps representing the backbone as a thin wire will help focusing on the important aspects.

Reviewer #2 (Remarks to the Author):

A lot of work has been done by the authors to answer the reviewers' comments in order to improve accuracy of some data and to improve the overall quality of the presented work. In my opinion, the results deserve now publication in Nature Comm.

Reviewer #3 (Remarks to the Author):

The authors have thoroughly addressed my concerns in their revised manuscript. I have one comment, which pertains to lines 109-117. The authors' important findings, which provide a structural basis for Chs1 activation by trypsin, are bolstered by the results of earlier genetic deletion analyses that indicated that the N-terminal region of the protein is dispensable for function. I'd encourage the authors to mention the following study: Ford, R. A., Shaw, J. A., Cabib, E., 1996 Yeast chitin synthases 1 and 2 consist of a non-homologous and dispensable N-terminal region and of a homologous moiety essential for function. *Mol. Gen. Genet.* 252: 420-428.

REVIEWER COMMENTS

Reviewer #1 (Remarks to the Author):

The revised manuscript by Dan-Dan Chen et al. has improved slightly, yet several important concerns have not been addressed.

Thanks for the valuable comments. We have performed more experiments and analysis to answer these questions as below.

Main concerns:

1. Chs1 activation: Prompted by the reviewers, the authors attempted to address the activation mechanism of Chs1. The observation that a swapping loop deleted Chs1 version exhibits significantly increased catalytic activity compared to the wild-type enzyme is intriguing. Now, the authors argue that swapping loop deletion renders the enzyme's N-terminal domain more prone to proteolysis and that its removal may account for activation. This idea is supported by the observed proteolysis of the wild-type enzyme and swapping loop mutant during purification.

To address this point, the authors generated an N-terminally truncated Chs1 construct equivalent to the remaining proteolytic fragment and show that this construct resists trypsin digestion. Accordingly, the authors conclude that the N-terminal domain is autoinhibitory and that its removal activates the enzyme. However, the critical experiment showing that the N-terminally truncated Chs1 version is catalytically active (without trypsin digestion) comparable to the swapping loop mutant or a trypsin digested wild-type enzyme is not shown. Hence, the conclusion that removal of the N-terminus activates Chs1 is not supported.

Thanks for the question. We have performed the activity assay of N-terminally truncated Chs1 (ΔN) following the suggestion, and found that the activity of ΔN is comparable to that of trypsin digested wild-type Chs1 (Fig. S3d). Combined with our earlier finding that trypsin treated Chs1 lacks the N-terminal region, we proposed proteolysis activation of Chs1 zymogen is by removing the N-terminal region.

It is also worth noting that the observed increase in activity of the trypsin digested and swapping loop deleted constructs (relative to full length) is not equivalent. Trypsin digestion appears to increase the wild-type activity about 3-fold (Fig. 1b), whereas swapping loop removal accounts for an increase of about 200-fold (Fig. 2e)

Thanks for this valuable question. As noted by the reviewer, the degrees of activation by swapping loop deletion is much higher than trypsin treatment (~20 fold (not 200 fold) vs ~3 fold). We agree that it's not logical to simply ascribe the increased activity of Chs1- Δ SL to proteolysis of the N-terminal inhibitory region.

During the revision, we first repeated the activity assay and double confirmed that the activity of Chs1- Δ SL was indeed more than 10 times that of our purified Chs1. Such high activity is unexpected because trypsin proteolysis only enhances the activity of Chs1 by 1.5-3 times as mentioned above in the manuscript. It means the increased activity by swapping loop deletion could not be only caused by the proteolysis of NTR (N-terminal region). Instead, it's likely that the swapping loop is directly involved in restricting the activity of Chs1, and plays the major inhibitory effect on Chs1 activity. This hypothesis is consistent with our later finding that GTD moves up and down in the catalytic cycle (**Fig. 3d**). The flexibility of swapping loop makes the movement of GTD possible.

As both proteolysis and swapping loop seems to be involved in the regulatory of activity, we further investigate their relation. We improved the map resolution of apo Chs1 from 3.0 to 2.6 Å during the revision. In the new map, densities around swapping loop was improved, and we were able to build two segments in NTR. The structure shows the segment 989-992 of swapping loop packs on segments 245-271 and 376-379 of NTR (**Fig. 2c, Fig. S14b**). Conceivably, removal of swapping loop will absolutely disrupt the interaction and make the NTR more flexible for proteolysis, which explains why purified Chs1- Δ SL was digested (**Fig. S14d**). Vice versa, the proteolysis of NTR will enhance the flexibility of swapping loop. It's possible that the proteolysis activation of Chs1 may actually be achieved by enhancing the flexibility of swapping loop, because removal of swapping loop have more activation effect on Chs1 activity. Indeed, the densities of the swapping loop and related two segments of NTR in our determined structures are relatively weaker than those in other parts of Chs1 or even invisible, indicating that they are only partially stabilized.

Further, the WGA chitin synthesis assays would benefit from a chitinase control reaction where chitin is specifically degraded prior to WGA binding.

Thanks for the advice. We performed the chitin degradation assay using purchased chitinase (C6137, Sigma). As shown in **Fig. S3e**, synthesized product by trypsin activated Chs1 (24h reaction) was visible as white precipitant and then specifically degraded by chitinase (24h incubation), suggesting the product's indeed chitin.

2. Substrate-bound states: In my opinion, the EM maps presented for UDP-GlcNAc in the absence and the presence of a priming monosaccharide are insufficient to be interpreted as UDP-GlcNAc molecule. Fig. S12 hardly shows any density for the donor sugar, perhaps with the exception of its acetamido group (although even this is not clear). This should not be interpreted as a donor-bound state and also not as a substrate-loaded

state that needs to rearrange for glycosyl transfer. Accordingly, the corresponding discussion of different binding poses of UDP versus UDP-GlcNAc ligands can either be removed or modified significantly, considering the uncertainty related to the presumed substrate-bound states.

Thanks for the comment. We agree the reviewer that densities of the GlcNAc moiety of UDP-GlcNAc in our earlier cryo-EM maps (UDP-GlcNAc bound; UDP-GlcNAc+GlcNAc bound) were relatively weaker than that of the UDP moiety.

To answer this question, we analyze the experiment carefully and noticed that densities of the GlcNAc moiety show well in lower threshold. We guess the weak density may be caused by contamination of UDP or partial hydrolysis of UDP-GlcNAc. Both of these problems lead to contaminated UDP bound in the active site and thus result in weaker density of the GlcNAc moiety in our maps. Therefore, we performed cryo-EM analysis on Chs1 in complex with UDP-GlcNAc or UDP-GlcNAc+GlcNAc using purer UDP-GlcNAc and less incubation time. With new data collection and processing, we obtained cryo-EM 3D maps of UDP-GlcNAc bound Chs1 at 3.1 Å resolution, and UDP-GlcNAc+GlcNAc bound Chs1 at 2.9 Å resolution. As shown in updated **Fig. S12**, the densities of the GlcNAc moiety of UDP-GlcNAc in our new cryo-EM maps are improved significantly. Modeling of the GlcNAc moiety in the new maps is very authentic. In these two new structures, the GlcNAc moiety of UDP-GlcNAc locate in similar position as in previous structures. All related figures were remade. We believe these new results can well support our conclusions.

3. Acceptor positioning: The authors modeled a GlcNAc monosaccharide at the acceptor binding site but do not discuss how this ligand is coordinated. Is its acetamido group resolved to allow proper positioning?

Thanks for the question. There are two Chs1 maps including the GlcNAc acceptor in our manuscript: the UDP-GlcNAc+GlcNAc bound and primed states at resolution of 3.14Å and 3.06Å respectively. In the primed Chs1 structure, the corresponding density of the GlcNAc acceptor is pretty good to model a monosaccharide in specific orientation, while the density in UDP-GlcNAc+GlcNAc bound is relative weaker. As mentioned above, we get an improved map of UDP-GlcNAc+GlcNAc bound Chs1 at 2.91Å resolution during the revision. Importantly, corresponding density of the GlcNAc acceptor in this map was also improved and a GlcNAc was able to be well modeled basing on the feature of its acetamido group (**Fig. S12**).

Interestingly, we found the GlcNAc in these two structures are likely in opposite orientations. It's reasonable to see GlcNAc in both orientations in the acceptor binding site, because two neighbor GlcNAc residues within the chitin chain are in alternating orientations. Due to the local resolution of corresponding densities are not high

enough to assign each atom of these two GlcNAc molecules, we think it would be better not to discuss specific ligand-residue interactions. Instead, we demonstrate the residue environment around the acceptor in page9: “At the bottom of the active site, Y654 in IF1, W760 in IF2, R718, and a conserved VLPGA motif (residues 673-677) form the GlcNAc acceptor binding site (Fig. 3a, b). The GlcNAc ring is vertically inserted into a slit between the indole ring of W760 and the pyrrolidine ring of P675, forming three parallel rings that are approximately 4 Å apart from each other. Below the GlcNAc molecule are a group of charged residues that keep GlcNAc in the binding site, such as E653, S657, and K662 of IF1.” Among these residues, importance of S657, K662, R718, W760, and the VLPGA motif were confirmed in later mutant assays.

4. Base catalyst: The authors refer to Asp602 as the base catalyst (line 244), yet according to the sequence alignment (Fig. S13), this residue belongs to a DAG motif presumably involved in cation/nucleotide binding. Could it be that Asp717 is the base catalyst?

We apologize for the typo and have revised the residue to D717.

5. Swapping loop modeling: I previously requested clarification on how the swapping loop was modeled, considering the poor map quality presented for this region. Based on the information presented in the rebuttal letter and the supplement, I cannot imagine how this loop could have been modeled to allow discussing individual side chain locations (lines 171-173). Perhaps a backbone model would suffice (or an AlphaFold model docked into the map, but this should be clearly stated)?

Sorry for not explaining clearly in our last revision. Indeed as the reviewer noted, our model was built using AlphaFold2 predicted structure as the initial model, and we have added this information in the main text (Page5 and 7).

Furthermore, we have collected new cryo-EM data of apo Chs1 and improved the map resolution from 3.0 to 2.6 Å during the revision. The density of swapping loop is improved and shows more side chain details in the new map (Fig. S14b).

6. Discussion, line 420-423: "...the following hydrolysis of the second donor molecule generates a free GlcNAc molecule, which then forms a disaccharide with the acceptor...". This sentence is almost certainly incorrect. GT-2 enzymes are believed to utilize an inverting SN2-like nucleophilic displacement mechanism, which means that the acceptor hydroxyl (C4 of the GlcNAc monosaccharide) attacks the C1 carbon of the donor sugar, such that its configuration is inverted from alpha to beta. Thus, the substrate molecule is not first hydrolyzed and then attached to the primer as this reaction requires an activated C1 carbon.

Thanks for the comment. We have revised the text to "the following hydrolysis of the second donor molecule and the formation of $\beta(1\rightarrow4)$ glycosidic bond with the acceptor GlcNAc generates a disaccharide."

Minor points:

Fig. 1b: It would be informative to test whether trypsin activated Chs1 is still sensitive to NikkoZ or PolyB inhibition.

Thanks for the suggestion. We performed the activity inhibition experiment of trypsin activated Chs1, and found its activity was also largely inhibited by NikkoZ and PolyB (Fig. S3d).

Fig. 4b and c: These panels are very difficult to interpret, certainly as a printed version. Perhaps representing the backbone as a thin wire will help focusing on the important aspects.

Thanks for the suggestion. We have remade the figures (Fig. 4b, c) by changing the backbone to thin wire and deepening the blue dots for better presentation.

Reviewer #2 (Remarks to the Author):

A lot of work has been done by the authors to answer the reviewers' comments in order to improve accuracy of some data and to improve the overall quality of the presented work. In my opinion, the results deserve now publication in Nature Comm.

We thank the reviewer for the affirmation of our work.

Reviewer #3 (Remarks to the Author):

The authors have thoroughly addressed my concerns in their revised manuscript. I have one comment, which pertains to lines 109-117. The authors' important findings, which provide a structural basis for Chs1 activation by trypsin, are bolstered by the results of earlier genetic deletion analyses that indicated that the N-terminal region of the protein is dispensable for function. I'd encourage the authors to mention the following study: Ford, R. A., Shaw, J. A., Cabib, E., 1996 Yeast chitin synthases 1 and 2 consist of a non-homologous and dispensable N-terminal region and of a homologous moiety essential for function. *Mol. Gen. Genet.* 252: 420-428.

Thanks for the valuable comments. We have added the citation in page 5: "This finding is also consistent with a previous study that deletion of the non-homologous N-terminal region of both Chs1 and Chs2 had little effect on the trypsin activated enzymatic activity of the corresponding synthase (ref 31)".

REVIEWERS' COMMENTS

Reviewer #1 (Remarks to the Author):

I think the authors have addressed my remaining concerns. In my mind, the revised manuscript is suitable for publication.

Yet, the authors should revise the following sentence of the discussion, lines 423-425:

'We further propose that the following hydrolysis of the second donor molecule and the formation of $\beta(1\rightarrow4)$ glycosidic bond with the acceptor GlcNAc generates a disaccharide.'

I suggest the following version:

'We further propose that a following substrate binding and turnover generates a $\beta(1\rightarrow4)$ glycosidic bond with the acceptor GlcNAc to form a disaccharide.'

The word 'hydrolysis' should be avoided because it implies a reaction with water. Instead, here the GlcNAc acceptor mediates the nucleophilic attack, not water.

REVIEWERS' COMMENTS

Reviewer #1 (Remarks to the Author):

I think the authors have addressed my remaining concerns. In my mind, the revised manuscript is suitable for publication.

We appreciate the reviewer's positive feedback of our work and the valuable comments.

Yet, the authors should revise the following sentence of the discussion, lines 423-425:

'We further propose that the following hydrolysis of the second donor molecule and the formation of $\beta(1\rightarrow4)$ glycosidic bond with the acceptor GlcNAc generates a disaccharide.'

I suggest the following version:

'We further propose that a following substrate binding and turnover generates a $\beta(1\rightarrow4)$ glycosidic bond with the acceptor GlcNAc to form a disaccharide.'

The word 'hydrolysis' should be avoided because it implies a reaction with water. Instead, here the GlcNAc acceptor mediates the nucleophilic attack, not water.

Thanks for the valuable comment. We have revised the sentence following the suggestion.